# Lidar Estimation of Rotor-Effective Wind Speed - An Experimental Comparison

Dominique P Held[1,2] and Jakob Mann[1]

[1]Department of Wind Energy, Technical University of Denmark (DTU), Frederiksborgvej 399, 4000 Roskilde, Denmark
[2]Windar Photonics A/S, Helgeshøj Alle 16-18, 2630 Taastrup, Denmark

*Correspondence to:* Dominique P Held (domhel@dtu.dk)

**Abstract.** Lidar systems have the potential of alleviating structural loads on wind turbines by providing a preview of the incoming wind field to the control system. For a collective pitch controller the important quantity of interest is the rotor-effective wind speed (REWS). In this study, we present a model of the coherence between the REWS and its estimate from continuous-wave nacelle-mounted lidar systems. The model uses the spectral tensor definition of the Mann model. Model results were compared to field data gathered from a 2- and 4-beam nacelle lidar mounted on a wind turbine. The comparison shows close agreement for the coherence and the data fits better to the proposed model than to a model based on the Kaimal turbulence model, which underestimates the coherence. Inflow conditions with larger length scales led to a higher coherence between REWS and lidar estimates than inflow turbulence of smaller length scale. When comparing the two lidar systems, it was shown that the 4-beam lidar is able to resolve small turbulent structures with a higher degree of coherence. Further, the advection speed by which the turbulent structures are transported from measurement to rotor plane can be estimated by 10 minute averages of the lidar estimation of REWS. The presented model can be used as a computationally efficient tool to optimize the position of the lidar focus points in order to maximize the coherence.

## 1 Introduction

The control system is an integral part of a wind turbine and has substantial influence on its behaviour. Its aim is to maximize the power production while keeping the turbine structural loading within the design limits. In order to decrease the levelized cost of energy, several novel sensors and control strategies have been proposed. One of them is a lidar-assisted pitch controller and one of the first was introduced by Harris et al. (2006). It utilizes nacelle- or spinner-mounted lidar systems to retrieve information about the inflow. In contrast to traditional feedback (FB) control of rotor speed, disturbances in the inflow can be measured by the lidar before they affect the turbine. For collective pitch control, a simple approach is to add a feedforward (FF) pitch angle demand $\beta_{FF}$ based on lidar measurements to the FB demand $\beta_{FB}$ derived from the rotor speed deviation from its desired value $\Omega_r$, see fig. 1.

For such a controller the important information about the wind is the rotor-effective wind speed (REWS) $v_{\text{eff}}$, which can be defined in several ways (Soltani et al., 2013). One definition states that the REWS is the average longitudinal wind speed component over the entire rotor plane, which is used in this work. Alternatively, the average of the longitudinal wind speed

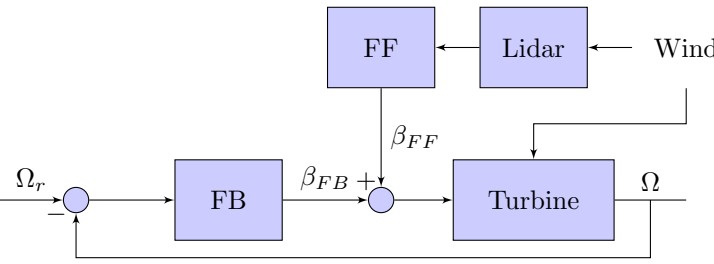

**Figure 1.** Block diagram of a lidar feedforward collective pitch controller to assist a traditional feedback controller.

with different weights for the hub or tip regions of the blade can be used. In the ideal case of perfect lidar measurements of $v_{\text{eff}}$ and turbine modeling, disturbances can be completely rejected and optimal rotor speed control can be achieved (Dunne et al., 2011). However, this is not achievable in reality and important shortcomings of the lidar systems are

- the contamination from lateral and vertical wind speed components,

- the spatial averaging due to the lidar's probe volume,

- the scarcity of measurement points in the rotor plane

- and the uncertain estimation of the time delay between lidar measurement and disturbance arrival at the rotor.

Thus, it is important to optimize the measurement positions of the lidar to maximize the correlation between lidar measurement and REWS for which flexible and computationally efficient models are required.

Previously, Schlipf et al. (2013) presented an analytic correlation model in frequency domain to calculate the magnitude-squared coherence and transfer function between a lidar and a turbine using the Kaimal spectral model and an empirical exponential decay coherence model of the longitudinal wind component for separations perpendicular to the flow. The turbulence model is defined in the IEC-61400-1 standard (IEC, 2005). The advantage of this approach compared to simulations in time domain is the reduced computational effort. However, integrating certain lidar properties becomes complicated if done

analytically. For example the spatial averaging effect of the lidar has not been integrated into the model. Therefore Schlipf et al. (2013) also proposed a semianalytic model, where properties of the lidar can be added in frequency domain and coherence and transfer function can then be calculated. The model has been extended by Haizmann et al. (2015a) to include linear rotor-effective horizontal and vertical shear estimation. Different optimal focus positions were found for REWS and shear estimations implying that a compromise needs to be found if both quantities are wanted. Another optimization was performed

in Schlipf et al. (2015), where additionally the wind evolution and constraints from the controller were considered. In Simley and Pao (2013b) a similar semianalytic method was presented to calculate the correlation between a spinner-mounted lidar and blade-effective wind speeds. The difference lies in the fact that spinner lidars rotate with the rotor and thus sample the wind field rotationally. In a more recent publication the theoretical and practical apsects of lidar optimization are addressed Simley et al. (2018). The results of the optimization of several lidar setups based on different optimization metrics are presented.

Comparisons between data gathered during field experiments and models were conducted in several studies. In Schlipf et al. (2013) the previously mentioned semianalytic model was compared against data gathered on NREL's CART2 test turbine. The measured and modeled transfer function showed very good agreement and the maximum coherent wavenumber, defined as the wave number where the coherence reaches a value of 0.5, was 0.06 rad/m for both methods. A similar comparison was performed on NREL's CART3 turbine by Scholbrock et al. (2013), where deviations between model and measured data was observed. As a possible explanation interference of the guy wires of a close-by meteorological mast with the lidar was given. In a later experiment on the same turbine with a different lidar system Haizmann et al. (2015b) found great agreement between data and model. For this lidar, the maximum coherent wavenumber was found to be 0.03 rad/m.

The integration of lidar measurements into turbine control by suitable controllers and their associated benefits have been the topic of various analyses. FF additions to FB controllers have been studied in e.g. Laks et al. (2011); Dunne et al. (2011). A more sophisticated flatness-based controller was proposed in Schlipf and Cheng (2014), while individual pitch controllers have been considered in e.g. Dunne et al. (2012). Model predictive control approaches were examined in e.g Mirzaei et al. (2013).

To verify simulated performances, field test have been pursued. In Scholbrock et al. (2013) a pulsed lidar system was used on NREL's CART3 turbine. A collective pitch feedforward approach (similar to fig. 1) was compared to a feedback controller only and load reduction at low frequencies (below 0.1 Hz) were observed. Damage equivalent loads (DELs) were reduced by approximately 2% and 7% for the tower fore-aft bending and the blade flapwise bending moments, respectively. A similar study was presented in Schlipf et al. (2014) using the CART2 turbine, where the blade and tower DELs were reduced by 10%. However, periods where the lidar's vision was obstructed by hard targets showed an increase in DELs and thus emphasizing the sensitivity of environmental conditions on lidar measurements. Another experiment on CART2 was performed by Kumar et al. (2015), where, besides adding a feedforward controller, the gains of the feedback controller has been reduced. Here the load analysis showed that a reduction was achieved after reducing the feedback gains.

In this paper we present a model of the coherence between REWS estimated from turbine and lidar measurements. The model uses the description of a turbulence field according to the model by Mann (1994), which allows to derive expressions of the auto- and cross-spectra numerically. In sec. 2 these expression are presented as well as the determination of REWS from field measurements at the turbine and from the lidar. Sec. 3 explains the test site and its characterization, while sec. 4 shows the measurement results and the comparison with the presented model. The model can be used as a computationally efficient tool to predict the auto- and cross-spectra of REWS from turbine and lidar measurements and the optimization of the lidar focus point positions.

## 2 Methodology

In this section we present a coherence model between nacelle lidar systems and a wind turbine. The theoretical expression to calculate the variances of turbine and lidar measurements have already been derived in Mirzaei and Mann (2016) and here we extend those to also calculate auto- and cross-spectra.

The fluctuating part of a three-dimensional (3D) wind field can be represented by the vector field $\boldsymbol{u}(\boldsymbol{x},t) = (u_1, u_2, u_3)$, where $\boldsymbol{x} = (x_1, x_2, x_3)$ and $\boldsymbol{k} = (k_1, k_2, k_3)$ refer to a 3D spatial and wavenumber domain, respectively. We assume that the vector field $\boldsymbol{u}(\boldsymbol{x},t)$ is frozen and the fluctuations are advected by the mean wind speed, i.e. Taylor's frozen turbulence hypothesis (Mizuno and Panofsky, 1975) applies:

$$\boldsymbol{u}(\boldsymbol{x},t) = \boldsymbol{u}(x_1 - Ut, x_2, x_3), \tag{1}$$

where $U = \langle \boldsymbol{u}(x_1, 0, 0) \rangle$ is the mean wind speed along the advection direction $x_1$. Thus, the dependence on time can be eliminated. The field $\boldsymbol{u}(\boldsymbol{x})$ can be written as a Fourier transform pair:

$$\boldsymbol{u}(\boldsymbol{x}) = \int \boldsymbol{u}(\boldsymbol{k}) e^{\mathrm{i}\boldsymbol{k}\cdot\boldsymbol{x}} d\boldsymbol{k} \iff \boldsymbol{u}(\boldsymbol{k}) = \frac{1}{(2\pi)^3} \int \boldsymbol{u}(\boldsymbol{x}) e^{-\mathrm{i}\boldsymbol{k}\cdot\boldsymbol{x}} d\boldsymbol{x}, \tag{2}$$

where an integral over the three-dimensional quantity, $\boldsymbol{k}$ or $\boldsymbol{x}$, means the integral from $-\infty$ to $\infty$ over all three components. The more rigorous Fourier-Stieltjes notation (Batchelor, 1953) was avoided due to brevity and clarity. The ensemble average of the absolute squared Fourier coefficients is the spectral tensor $\Phi_{ij}(\boldsymbol{k})$

$$\langle u_i^*(\boldsymbol{k}) u_j(\boldsymbol{k}') \rangle = \Phi_{ij}(\boldsymbol{k}) \delta(\boldsymbol{k} - \boldsymbol{k}'), \tag{3}$$

where $\delta(.)$ is the Dirac delta function. Since $\boldsymbol{u}^*(\boldsymbol{k}) = \boldsymbol{u}(-\boldsymbol{k})$, eq. 3 can be written as

$$\langle u_i(\boldsymbol{k}) u_j(\boldsymbol{k}') \rangle = \Phi_{ij}(\boldsymbol{k}) \delta(\boldsymbol{k} + \boldsymbol{k}'). \tag{4}$$

The advantage of using a three-dimensional spectral tensor $\Phi_{ij}$ compared to one-dimensional spectra and coherences as in Schlipf et al. (2013) is that the problem of comparing lidar derived wind speed estimates with rotor-averaged winds is a truly three-dimensional question. As we will see later, the lidar beams are focused at different locations measuring different velocity components and the resulting spectra can be naturally expressed as weighted two-dimensional integrals over the three-dimensional spectral tensor, see for example eq. 22. In this paper we use the spectral tensor model by Mann (1994) which is given analytically and only contains three adjustable parameters: $\alpha_K \epsilon^{2/3}$, $L$, and $\Gamma$. The first is the spectral Kolmogorov constant $\alpha_K$ (Monin and Yaglom, 1975) multiplied with the dissipation rate of specific turbulent kinetic energy density $\epsilon$ to the two-thirds power. This parameter gives the level on the spectra in the inertial subrange (Kolmogorov, 1941). The second parameter $L$ is a length scale describing the size of the eddies containing most energy. Finally, $\Gamma$ is a parameter describing the deviation from isotropy which is caused by the mean shear usually present in the lower parts of the atmosphere. When $\Gamma = 0$ turbulence is isotropic while typical values in the atmospheric surface layer where most wind turbines are present is between 3 and 4 (Peña et al., 2010; Sathe et al., 2012; de Maré and Mann, 2014; Chougule et al., 2015). Even though the spectral tensor of Mann is given analytically, the usual one-dimensional spectra which are the ones measured by, for example, sonic anemometers, cannot be expressed in closed form but have to be numerically integrated[1]. This is where the Kaimal model has an advantage as the one-dimensional spectra are given as simple analytic expressions (Kaimal and Finnigan, 1994; IEC, 2005). However, each of the three velocity component spectra are described by two parameters (a length scale and a variance) making in total

---

[1]A look up table of one-dimensional spectra obtained from the Mann model can be obtained from the second author at `jmsq@dtu.dk`

six parameters and on top of that all coherences between velocity components also have to be specified. In the IEC standard only the coherence of the longitudinal velocity is specified. That requires three new parameters and if all coherences for all combinations of velocity components were to be described in the same way, which is necessary for the present investigations, the number of parameters would be overwhelming. Additionally, the Mann model is related to the physical equations of the flow, i.e. the continuity equation and a linearized version of the Navier-Stokes equations describing the effect of the shear on the turbulence. With these advantages we feel confident that using the Mann model for these investigations is a sensible choice.

## 2.1 Overview Coherence Model

In this study we have used an estimation of the coherence to evaluate the correlation between models and measurements. Specifically, we were interested in the magnitude squared coherence between the REWS measured at the turbine and estimated from lidar measurements

$$\gamma_{\mathrm{RL}}(k_1) = \frac{|S_{\mathrm{RL}}(k_1)|^2}{S_{\mathrm{LL}}(k_1)S_{\mathrm{RR}}(k_1)}, \tag{5}$$

where $S_{\mathrm{LL}}$ and $S_{\mathrm{RR}}$ are the auto-spectra of the lidar and turbine estimates of REWS and $S_{\mathrm{RL}}$ is their cross-spectrum. From time series measurements these spectra were calculated over a 10 minute period. The resulting frequency domain is converted into wavenumber domain by the use of Taylor's frozen turbulence hypothesis using $k_1 = \frac{2\pi f}{U}$. The wavenumber describes the size of a turbulence fluctuation, where a small wave number indicates a (spatially) large fluctuation and vice versa. The remainder of this section will present the methods to calculate the REWS and spectra. At the end the model is compared against numerical simulations to validate the implementation.

## 2.2 REWS estimated from turbine measurements

The REWS $v_{\mathrm{eff}}$ is the defined as the longitudinal wind vector component averaged over the entire rotor plane

$$v_{\mathrm{eff}}(x_1) = \frac{1}{\pi R^2} \iint_{\mathrm{rotor}} u_1(\boldsymbol{x})dx_2 dx_3 = \frac{1}{\pi R^2} \iint_{\mathrm{rotor}} \int u_1(\boldsymbol{k})e^{i\boldsymbol{k}\cdot\boldsymbol{x}}d\boldsymbol{k}dx_2 dx_3 \tag{6}$$

$$= \frac{1}{\pi R^2} \int u_1(\boldsymbol{k})e^{ik_1 x_1} \iint_{\mathrm{rotor}} e^{i(k_2 x_2 + k_3 x_3)}dx_2 dx_3 d\boldsymbol{k} \tag{7}$$

$$= \int u_1(\boldsymbol{k})e^{ik_1 x_1} \frac{2J_1(\kappa R)}{\kappa R}d\boldsymbol{k}, \tag{8}$$

where $R$ is the rotor radius, $\kappa = \sqrt{k_2^2 + k_3^2}$ and $J_1$ is the Bessel function of the first kind and where the Fourier transform of the velocity field has been introduced by eq. 2. The rotor is positioned perpendicular to the $x_1$-axis, i.e. no yaw misalignment.

To calculate the auto-spectrum of $v_{\mathrm{eff}}$ we first work out the auto-correlation using eq. 8:

$$R_{\mathrm{RR}}(x_1) = \langle v_{\mathrm{eff}}(x_1')v_{\mathrm{eff}}(x_1' + x_1) \rangle \tag{9}$$

$$= \left\langle \int u_1^*(\boldsymbol{k})e^{-ik_1 x_1'} \frac{2J_1(\kappa R)}{\kappa R}d\boldsymbol{k} \int u_1(\boldsymbol{k}')e^{ik_1'(x_1' + x_1)} \frac{2J_1(\kappa' R)}{\kappa' R}d\boldsymbol{k}' \right\rangle . \tag{10}$$

Notice that we have complex conjugated the first term which is allowable because it is real. Each complex term in the integral is therefore conjugated. We now change the product of integrals into a double integral and move the ensemble averaging inside this integral. Since the Fourier transforms of the velocities are the only stochastic variables in the expression the ensemble average can be moved so it only embraces the product of $u_1^*(\boldsymbol{k})$ and $u_1(\boldsymbol{k}')$. Now we use eq. 3 to introduce the spectral tensor and performing the integral over $\boldsymbol{k}'$ leaves us with a single three-dimensional integral:

$$R_{\text{RR}}(x_1) = \int \Phi_{11}(\boldsymbol{k}) \frac{4J_1^2(\kappa R)}{\kappa^2 R^2} e^{ik_1 x_1} d\boldsymbol{k} \tag{11}$$

This equation is now inserted into the definition of the spectrum

$$S_{\text{RR}}(x_1) = \frac{1}{2\pi} \int R_{\text{RR}}(x_1) e^{-ik_1 x_1} dx_1 \tag{12}$$

and the Fourier transform essentially annihilates the integral over $k_1$ in eq. 11 and we are left with the final two-dimensional integral expression for the auto-spectrum of $v_{\text{eff}}$

$$S_{\text{RR}}(k_1) = \iint\limits_{-\infty}^{\infty} \Phi_{11}(\boldsymbol{k}) \frac{4J_1^2(\kappa R)}{\kappa^2 R^2} dk_2 dk_3. \tag{13}$$

To estimate $v_{\text{eff}}$ from signals measured on the turbine the approach in Østergaard et al. (2007) was followed. It is based on using the entire rotor as an anemometer and derive the rotor-effective wind speed by considering the turbine model characteristics and several measured signals. The methods gives the magnitude of an undisturbed wind field that creates the (unique) combination of power production, rotational speed and pitch angle at the turbine. Thus, there is no need to correct for the effect of turbine induction.

The entire turbine is modelled by a simple drive train model

$$J\dot{\Omega} = Q_a - Q_g - Q_{\text{loss}}, \tag{14}$$

where $J$ is the moment of inertia of the drive train, $\Omega$ is the rotational speed of the rotor, $Q_a$ is the aerodynamic torque produced by the rotor, $Q_g$ is the generator torque and $Q_{\text{loss}}$ is a collective term for the lost torque along the drive train. In our field experiment, torque measurements at the low-speed shaft (LSS) were performed. Thus, the measurements are taken before the gearbox and generator (where most of the losses occur) and we can replace $Q_{\text{LSS}} = Q_g + Q_{\text{loss}}$ in eq. 14. The sampling rate of the turbine data was 1 Hz. Further a low-pass filter was used to reduce the influence of measurement noise in the estimation of $\dot{\Omega}$. The aerodynamic torque is defined by

$$Q_a = \frac{1}{2}\rho\pi R^2 \frac{v_{\text{eff}}^3}{\Omega} C_p(\beta, \lambda) = \frac{1}{2}\rho\pi R^2 \frac{R^3 \Omega^2}{\lambda^3} C_p(\beta, \lambda), \tag{15}$$

where $\rho$ is the air density, $\lambda = \frac{\Omega R}{v_{\text{eff}}}$ is the tip-speed ratio (TSR), $C_p(\beta, \lambda)$ is the power coefficient as function of pitch angle $\beta$ and TSR. By solving eq. 14 for $Q_a$ and substitute it in eq. 15 we arrive at

$$\frac{C_p(\beta, \lambda)}{\lambda^3} = \frac{2Q_a}{\rho\pi R^5 \Omega^2} = \frac{2(Q_{\text{LSS}} + J\dot{\Omega})}{\rho\pi R^5 \Omega^2}. \tag{16}$$

With a measurement of the pitch angles, a look-up with linear interpolation can be used to find $\lambda$ that satisfies eq. 16 and by the definition of the TSR the REWS can be estimated

$$\hat{v}_{\text{eff,R}} = \frac{\Omega R}{\lambda}. \tag{17}$$

The necessary $C_p(\beta, \lambda)$ surface can be precomputed. Details can be found in appendix A. Note that issues of non-monotonicity of $C_p(\beta, \lambda)$ in eq. 16 can be avoided by performing the look-up on $\frac{C_p(\beta, \lambda)}{\lambda^3}$ and not on $C_p(\beta, \lambda)$. The air density has been calculated from pressure and temperature measurements on a nearby meteorological mast.

## 2.3 REWS estimated from lidar measurements

Lidar systems are able to measure the frequency shift of light back-scattered at aerosols moving with the wind in the atmosphere. This frequency shift is proportional to the velocity of the aerosols and thus the wind speed can be determined. The measurement of a continuous-wave lidar system can be expressed mathematically as the convolution of the line-of-sight (LOS) component of the wind vector and a weighting function given by the laser light intensity along the laser beam:

$$v_{\text{LOS}}(\boldsymbol{x}_f) = \int\limits_{-\infty}^{\infty} \boldsymbol{n} \cdot \boldsymbol{u}(s\boldsymbol{n} + \boldsymbol{x}_f)\varphi(s - d_f)ds, \tag{18}$$

where $\boldsymbol{x}_f$ is the position of the lidar focus point, $\boldsymbol{n}$ is the laser beam unit vector,

$$\varphi(s) = \frac{1}{\pi}\frac{z_R}{z_R^2 + s^2} \tag{19}$$

is the weighting function defined by the Rayleigh length $z_R$ and $d_f$ is the distance from the lidar system to the focus point. The Rayleigh length describes the shape of the probe volume through eq. 19. Note that the probe volume of the lidar increases with focus distance, i.e. $z_R \propto d_f^2$. The probe volume has an attenuating effect on the turbulent fluctuations of the wind field. Eq. 18 is assuming that the first statistical moment is used to calculate the dominant frequency of the Doppler spectrum, which is a result of the Fourier analysis of the detected light signal. Different frequency estimators can yield less turbulence attenuation (Held and Mann, 2018). The Fourier transformation of the weighting function (eq. 19) is $\mathcal{F}[\varphi(s)](k) = \exp(-z_R|k|)$ and the auto-spectrum of the lidar measurement along a single beam can be expressed as

$$S_{\text{LL}}(k_1) = n_i n_j \iint\limits_{-\infty}^{\infty} \Phi_{ij}(\boldsymbol{k})e^{-2z_R|\boldsymbol{k} \cdot \boldsymbol{n}|}dk_2 dk_3, \qquad \text{(one beam only)} \tag{20}$$

where $n_i$ refers to the components of the laser beam unit vector $\boldsymbol{n}$ and summation of repeated indices is implied. The implied sum $n_i \Phi_{ij} n_j \equiv \sum_{i=1}^{3} \sum_{j=1}^{3} n_i \Phi_{ij} n_j$ could also be written in vector and matrix notation as $\boldsymbol{n} \cdot (\Phi \boldsymbol{n})$ where the parenthesis is a product between a matrix and a vector r and a $\cdot$ means the dot product. Details of the derivation can be found in Mirzaei and Mann (2016).

The typical setup of a nacelle lidar looking forward is shown in the left part of fig. 4. The lidar systems probes sequentially several focus points in front of the rotor. Due to the limitation of measuring only the LOS component of the wind vector

assumptions are necessary to derive the REWS from the lidar measurements. Here we apply the following assumptions: (1) no vertical components, (2) zero turbine yaw misalignment. Based on these assumptions the REWS can be estimated from lidar measurements as the average of all $v_{\mathrm{LOS}}$ velocities:

$$v_{\mathrm{eff,L}} = \frac{1}{b \cos \alpha} \sum_{i=1}^{b} v_{\mathrm{LOS},i}, \tag{21}$$

where $b$ is the number of beams and $\alpha$ is the half-cone opening angle of the scanning cone (see left panel of fig. 4).

In wave-number domain the auto-spectrum of the REWS estimate from lidar measurement using eq. 21 and 18 can be written as:

$$S_{\mathrm{LL}}(k_1) = \frac{1}{b^2 \cos^2 \alpha} \sum_{i,j=1}^{b} \iint\limits_{-\infty}^{\infty} n_{ik} \Phi_{kl}(\boldsymbol{k}) n_{jl} e^{\mathrm{i} d_f \boldsymbol{k} \cdot (\boldsymbol{n}_i - \boldsymbol{n}_j) - z_R (|\boldsymbol{k} \cdot \boldsymbol{n}_i| + |\boldsymbol{k} \cdot \boldsymbol{n}_j|)} dk_2 dk_3 \tag{22}$$

where the index $k$ should not be confused with the wavenumber $k$. The details of the derivation can be found in Mirzaei and Mann (2016) with the only difference that they integrate over $k_1$ to get the variance whereas we do not in order to get the one-dimensional spectrum.

When evaluating eq. 21 from field measurement a correction of the slowdown in speed as the wind approaches the turbine is necessary. This correction is referred to as induction correction. First, we used an analytic solution for the flow speed reduction and diversion around the rotor (Conway, 1995). The model assumes an actuator disk model and laminar, uniform inflow with uniform, non-rotational loading. An example of the flow around a rotor can be found in the appendix B. The induction correction $a_c = \frac{U}{U_\infty}$, where $U_\infty$ is the undisturbed free steam wind speed, can be defined from the calculated flow field at the focus positions of the lidar beams. Thus, the induction correction depends on lidar parameters, i.e. the half-cone opening angle $\alpha$ and the focus distance $d_f$, and on the operational point of the turbine, i.e. the induction factor $a$. The induction factor is determined from the measured 10 minute mean REWS by the lidar and a steady-state thrust curve is used to look up the thrust coefficient $C_t$. Then the relation $C_t = 4a(1-a)$ is used to calculate the induction factor. The effect of the induction is assumed to be constant over a 10 minute period.

Integrating the induction corrections into eq. 21 yields

$$\hat{v}_{\mathrm{eff,L}} = \frac{1}{a_c} \frac{1}{b \cos \alpha} \sum_{i=1}^{b} v_{\mathrm{LOS},i}. \tag{23}$$

This represents measuring the wind speed component perpendicular to the turbine rotor even when the turbine is misaligned to the free-stream wind direction. A turbulent structure traveling along the mean wind direction, which is not aligned with the rotor axis if yaw misalignment is present, can arrive at the different position at the rotor than predicted by the model. The model assumes that the turbulent structures travel along the mean wind direction and that the turbine is aligned to that wind direction. However, in case of small yaw misalignment the shortcoming of the model can be assumed to be insignificant.

## 2.4 Determination of the cross-spectrum

Similarly the cross-spectrum between the REWS and its estimate from lidar measurements $v_{\text{eff,L}}$ can be calculated using

$$S_{\text{RL}}(k_1) = \frac{1}{b \cos \alpha} \sum_{i=1}^{b} \iint_{-\infty}^{\infty} n_{ij} \Phi_{j1}(\boldsymbol{k}) e^{\text{i}(d_f \boldsymbol{k} \cdot \boldsymbol{n}_i + k_1 \Delta x)} e^{-z_R |\boldsymbol{k} \cdot \boldsymbol{n}_i|} \frac{2J_1(\kappa R)}{\kappa R} dk_2 dk_3, \tag{24}$$

where $\Delta x$ indicates the distance between rotor and lidar measurement plane. Details of the derivation of the previous two equations can be found in Mirzaei and Mann (2016). Note that it is assumed that the lidar system probes focus points simultaneously, which is a simplification of the sequential scanning performed by the actual lidar system.

## 2.5 Model implementation and validation against simulations

For the implementation of the model a C++ code has been created to numerically solve eq. 13, 22 and 24. Adaptive cubature integration was used as an integration algorithm[2]. To validate the implementation numerical simulations have been performed. At first six random 3D turbulence boxes with different turbulence seeds have been created according to the Mann spectral tensor (Mann, 1998)[3]. The boxes had dimensions of 2800 m x 64 m x 64 m using 8192 x 32 x 32 grid points per box and contained only the turbulent part of the wind field, i.e. the mean wind speed was zero. The lidar measurements have been simulated using eq. 18, however due to the finite size of the boxes the integration has been truncated at $\pm 10 z_R$ from the focus point; details can be found in Held and Mann (2018). The two lidar systems presented in tab. 1 have been used. The rotor plane (with a diameter of 52 m) was discretized by 100 x 100 grid points.

The results of the coherence analysis can be found in fig. 2. It can be seen that the coherence for the 2-beam lidar drops at lower wavenumbers than the 4-beam lidar. This is due to the greater coverage of the rotor plane using four distinct focus locations compared to only two for the 2-beam lidar. Further, the comparison between the simulations and the model shows very good agreement. Some deviations remain, which can be attributed to using only six simulations when estimating the coherence.

## 3 Experimental setup

### 3.1 Instrumentation

Field measurements have been conducted at DTU's test site at Risø, located at the Roskilde Fjord in Denmark. The site consists of one row of wind turbines intended for testing and several meteorological masts are installed around the turbines, see fig. 3. During the experiments only a Nordtank was operative, which is located at a distance of 215 m $(4.1D)$ at an angle of $195°$ (from north). In general, there is a slight positive terrain slope from the fjord towards the turbines. To the east of the Vestas V52 some buildings and vegetation exists, while towards the west the turbine is facing flat fields and the fjord.

---

[2]The adaptive cubature integration scheme was written by Steven G. Johnson and is available on GitHub: https://github.com/stevengj/cubature

[3]The software can be downloaded free of charge at http://www.wasp.dk/weng#details__iec-turbulence-simulator

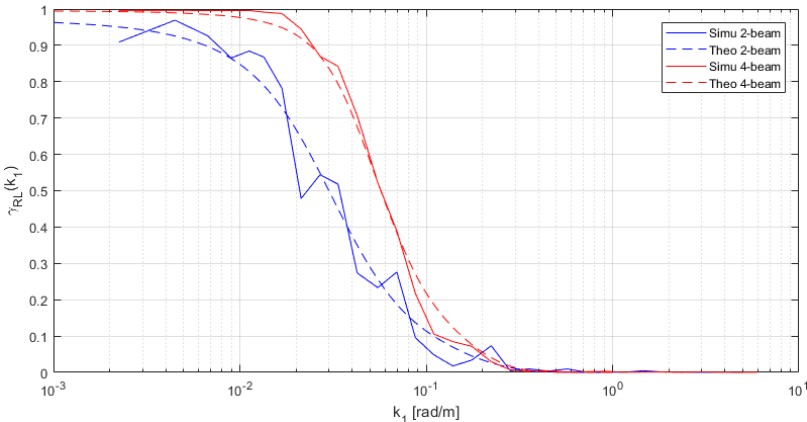

**Figure 2.** Coherence between the estimation of REWS from the turbine and the lidar. The comparison between the numerical simulations (*Simu*) and the implementation of the model (*Theo*) show very good agreement.

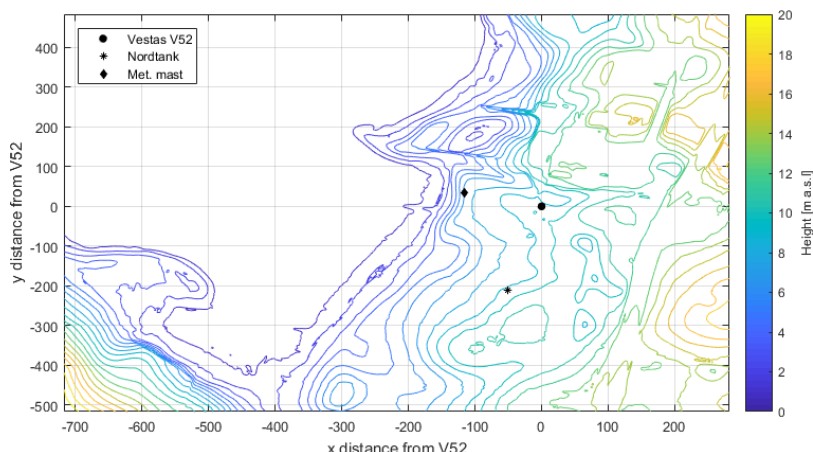

**Figure 3.** Digital terrain model (DTM) of the DTU's test site at Risø, where the Vesats V52, its meteorological mast and the Nordtank turbine are indicated. Zone 32 UTM coordinates centered at the Vestas V52 turbine were used. The DTM data was obtained from the Danish Map Supply (Agency for Data Supply and Efficiency, 2018). The units on both axes are meters.

For this experiment two continuous-wave coherent Doppler lidars manufactured by Windar Photonics A/S have been mounted on a Vestas V52 turbine. The lidar systems, a 2-beam and a 4-beam lidar, are mounted on the nacelle of the turbine and have been staring forwards to measure the inflow of the turbine. An illustration and a photo of the 4-beam lidar can be seen in fig. 4. The specifications for both lidars can be found in tab. 1. The two systems both contain one laser source located

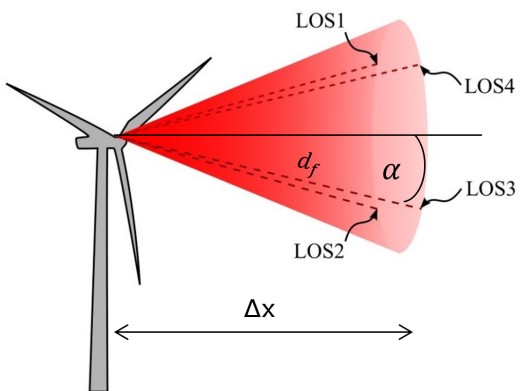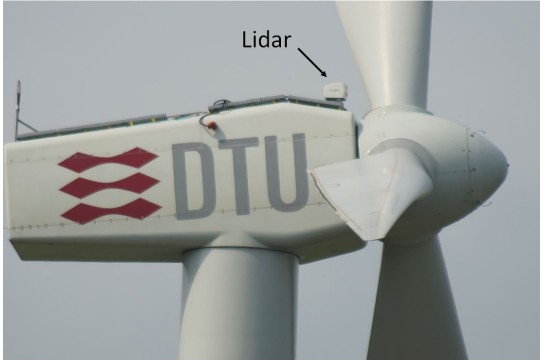

**Figure 4.** *Left*: Illustration of the 4-beam lidar focus locations on a cone with apex at the turbine nacelle. *Right*: Photo of the 4-beam lidar mounted on the Vestas V52 turbine at the Risø test site.

5    inside the nacelle and switch between the focus point sequentially. Each scan is completed in one second. Note the different Rayleigh lengths due to the different focus distances and the increased half-cone opening angle for the 2-beam system. The azimuth angle refers to the position on the scanning cone surface. The position at the top of the cone is at an azimuth angle of $0°$. Hence, the 2-beam lidar consists of two horizontal beams, while the 4-beam lidar has one focus point in each quadrant of the rotor area. The distance between the lidar system and the rotor is $d_{\mathrm{Nac}}$.

**Table 1.** Information of lidar setup parameters and measurement periods. The azimuth angle refers to the position on the scanning cone surface with $0°$ being the top of the cone.

|  | 2-beam | 4-beam |
|---|---|---|
| Focus distance $d_f$ [m] | 37 | 62 |
| Rayleigh length $z_R$ [m] | 2.1 | 6.0 |
| Half-cone opening angle $\alpha$ [°] | 30 | 18 |
| Azimuth angle [°] | 90 and 270 | 45, 135, 225 and 315 |
| Distance focus points - rotor $\Delta x$ [m] | 32 | 59 |
| Distance lidar - rotor $d_{\mathrm{Nac}}$ [m] | $\approx 1$ | $\approx 1$ |
| Scan time per beam [s] | 0.5 | 0.25 |
| Sampling rate [Hz] | 1 | 1 |
| Period measured | 30th Mar – 3rd May 2016 | 21st Oct – 15th Dec 2016 |

The Vestas V52 turbine has a diameter of 52 m and a hub height of 44 m with a rated power of 850 kW. It is heavily instrumented with several mechanical strain gauges, in particular a strain gauge set-up to measure the torque on the low-speed shaft. Also a meteorological mast is located approximately $2.5D$ in front of it. To characterize the flow conditions during the experiment a Metek P2901 USA-1 3D sonic anemometer mounted at hub height was used. Further measurements from a Vaisala PTB110 air pressure sensor and a Vaisala R/H HMP 155 humidity sensor were used in addition to the temperature measurements from the sonic anemometer to calculate the air density.

## 3.2 Site characterization

The wind rose derived from wind direction and horizontal wind speed of the sonic anemometer measurements during the periods of the experiment are presented in the left panel of fig. 5. The main wind direction is from the west with winds coming from the Roskilde Fjord.

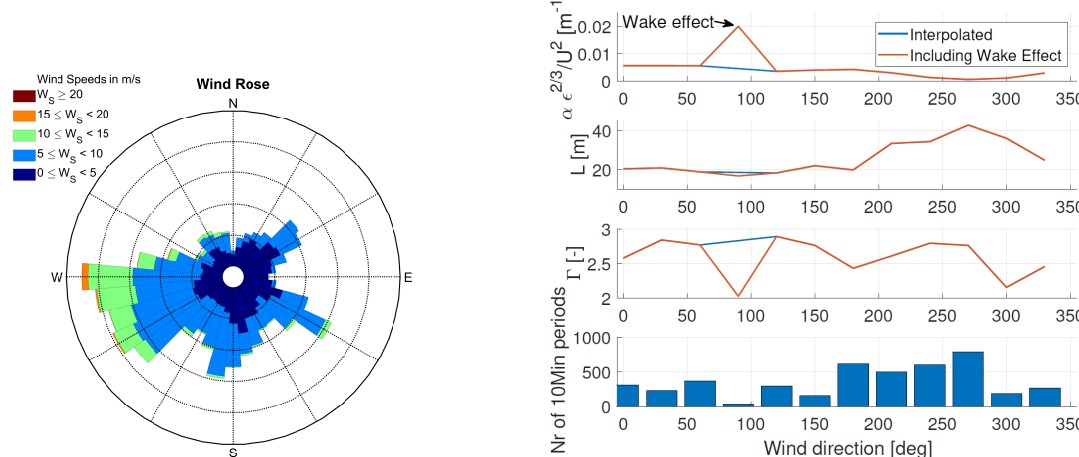

**Figure 5.** *Left*: Wind rose gathered during the test periods at the Risø test site. *Right*: The three top panels show the result of fitting the Mann Model to the calculated mean spectra as function of wind direction. The bottom panel shows the number of acquired 10 minute periods as function of wind direction.

To get a clearer picture of the inflow conditions the data set was grouped into sectors of $30°$ and the Mann model has been fitted to the average spectra in each sector. The fitting followed the procedure in Mann (1994) and was performed on the $u$-, $v$-, $w$-spectra and the $uw$-co-spectra. The spectra have been normalized by the mean wind speed squared. The model has three parameters: $\alpha_K \epsilon^{2/3}$, $L$ is a length scale and $\Gamma$ is an anisotropy parameter as already explained in section 2, for further details see Chougule et al. (2015). The results of the three model parameter as function of wind direction can be seen in the right panel of fig. 5. First of all, the effect of the wake from the Vestas V52 turbine onto the sonic anemometer is clearly seen at a wind direction of $90°$. In this sector a very high turbulent kinetic energy dissipation rate and a low anisotropy parameter were calculated. The results from this sector were disregarded and linearly interpolated. Secondly, two wind regimes can be identified. A region spanning from $330°$ to $180°$ shows a length scale $L$ of approximately 20 m, while for the region from

$210°$ - $300°$ larger length scales were fitted. Similarly the normalized dissipation rate is higher in the first region compared to the second. This is in agreement with the terrain of the test site. The inflow for the first region is characterized by obstacles like buildings and tall vegetation. The second region faces open fields and the fjord fetch. The fit has also been performed for the Kaimal turbulence model, which is defined in the IEC 61400-1 standard (IEC, 2005). This model has one characteristic length parameter $L_k$. Here only the $u$-, $v$- and $w$- spectra were fitted. The measured spectra have been been normalized by their measured variance and the frequency domain has been converted into wave number domain. Then the model was fitted to the measured spectra by minimizing the combined mean squared error of the spectra.

We separated the following analysis into two regions. The information on the two regions can be found in tab. 2 including the averaged fits to the Mann model. Note that these two regions do not refer to the operational regions of the controller. More 10 minute periods were obtained for region 2 due to the dominant wind direction from the west. The fitting results for the Kaimal model can also be found in tab. 2. Similar to the Mann model a larger length scale parameter was found for region 2. In Appendix C the spectra and the fitted Mann model can be found. By dividing the data into two regions, as shown in tab. 2, it was possible to identify two wind regimes with different turbulence parameter. It was also observed that binning according to atmospheric stability did not lead to significantly different spectra.

**Table 2.** Measurement sectors and fitted Mann model ($\frac{\alpha\epsilon^{2/3}}{U^2}$, $L$ and $\Gamma$) and Kaimal model ($L_k$) parameter.

|  | Region 1 | Region 2 |
|---|---|---|
| Direction | $330°$ - $180°$ | $210°$ - $300°$ |
| Nr. of 10-min periods | 1678 | 2713 |
| $\frac{\alpha\epsilon^{2/3}}{U^2}$ [$10^{-3}$ m$^{-1}$] | 4.29 | 1.60 |
| $L$ [m] | 18.5 | 37.9 |
| $\Gamma$ [-] | 2.36 | 2.41 |
| $L_k$ [m] | 201.0 | 326.6 |

# 4 Results

The first step in the analysis of the results was to apply appropriate data filters, which reject measurements based on certain criteria. It was necessary to identify periods where the turbine was in a normal power production state. Thus, a lower threshold on the minimum power production (i.e. $>0\,$kW), minimum rotor speed (i.e. $>16\,$rpm) and maximum pitch angle (i.e. $<23°$) in a 10 minute period were utilized. These thresholds were found by inspection of the available turbine data. This filter removed 52.8% and 53.8% of the data for the 2- and 4-beam experiments, respectively.

The filter applied to the lidar data consisted of a minimum number ($>90$% or 540 measurements) of available measurements on each beam in a 10 minute interval. Unavailable measurement have been interpolated linearly. Instances where 4 or more consecutive unavailable measurements occurred on any beam were also discarded. Whether a measurement is available or not was decided internally by the lidar system and depends on carrier-to-noise ratio and the Doppler peak shape and area. After

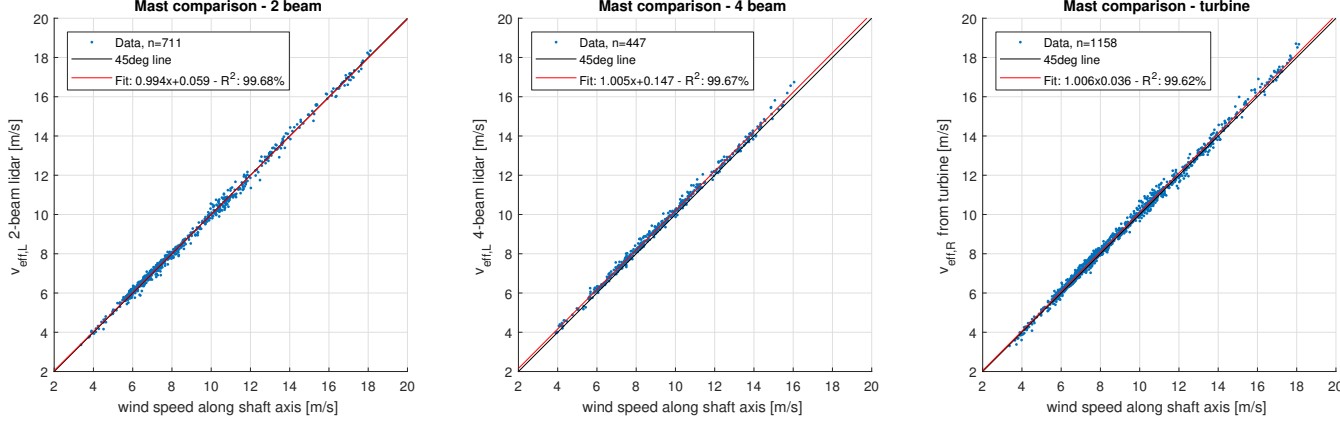

**Figure 6.** Comparison of 10-minute REWS estimates from of the lidars and the turbine to the meteorological mast's sonic anemometer. The data was taken from periods when the turbine was operational and facing the mast.

applying the turbine availability filter, this filter for the lidar data lead to an additional discard of 2.9% and 15.3% for the 2- and 4-beam system, respectively. Since the 4-beam system was under development during the field test a higher unavailability is observed. Note that the interference by the blades is removed by the lidars systems based on the Doppler spectra, which have a distinct shape when laser light is reflected at the blades.

Additionally, inflow from all yaw position except from the wake sector ($195° \pm 30°$) were considered because the lidar yaw misalignment measurements are biased in wake situations (Held et al., 2018). The yaw position filter lead to an additional exclusion of 6.0% and 6.3% of the data for the 2- and 4-beam periods, respectively.

### 4.1 Comparison of mean wind speeds

Next, the 10 minute average REWS estimates of lidar and turbine are compared to the sonic anemometer mounted on the meteorological mast. The comparisons for the 2 lidar systems and the turbine can be found in fig. 6. Besides the previously mentioned data filters, only yaw positions, where the turbine was facing the meteorological mast (i.e. a yaw heading of $289° \pm 20°$) have been considered. It can be seen that the both lidar systems agree well with the mast's sonic anemometers; linear least-square fitting results in a slope close to unity with no significant bias. This indicates that the correction for turbine misalignment and induction are working as intended. Similarly, the correlation between mast and turbine also shows very good accordance with no systematic error.

Corresponding comparisons are performed between the REWS estimated from the lidar systems and the turbine, respectively. The correlation plots can be found in fig. 7. Analogous to the comparisons to the mast, theses comparisons also show that there is no systematic error between the two signals. Both linear fits show unity slopes and very small offsets. They are slightly worse than the comparisons to the meteorological mast, which can be explained by model inaccuracies when estimating the REWS when using turbine and lidar data.

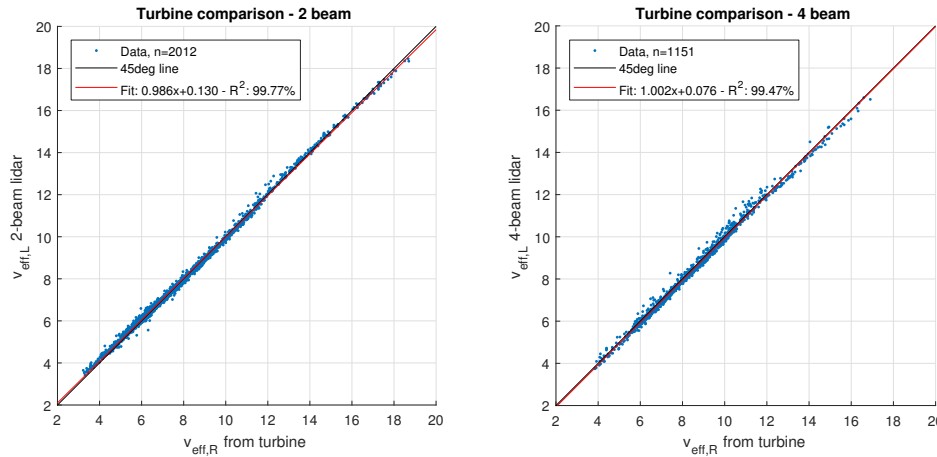

**Figure 7.** Comparison of 10 minute REWS estimates between the lidars and the turbine. Data from the wake sector and nonoperational periods of the turbine were removed. All units are meter/second.

For illustrative purposes, the next plots in fig. 8 show a single time series result for the lidar and turbine estimate of REWS. Both signals have a sampling rate of 1 Hz. In general, it can be seen that the fluctuations in REWS that were sensed by the rotor can also be measured by the lidar systems. For the 2-beam lidar larger deviations can be observed since this lidar probes the incoming wind field only at two locations. Fluctuations that occur at the top or bottom parts of the rotor can not be measured. In case of the 4-beam system a measurement in each quadrant is performed and gives a better estimate of the wind speed effecting the entire rotor. Further, the preview ability of the lidar systems also becomes apparent. Fluctuations can be measured before they affect the rotor.

## 4.2 Comparison of coherence

The effect of probing two versus four focus locations is now studied in wavenumber domain by comparing the squared coherences. The experimental data is also compared to two models: the model based on the Mann turbulence model introduced in sec. 2 and the Kaimal turbulence model used in previous studies (Schlipf et al., 2013). As mentioned in sec. 3.2, the analysis was split into two regions, of which one is disturbed by buildings or trees (*region 1*) and the other has an undisturbed inflow over open fields or the fjord's fetch (*region 2*).

The coherence analysis for region 1 can be found in fig. 9. At first, it can be seen that the coherence of the 2-beam lidar drops at lower wavenumbers than the coherence of the 4-beam lidar indicating that small fluctuations can be sensed more accurately by the 4-beam system. Secondly, the measured data agrees very well with the Mann turbulence model coherence. The Kaimal model on the other hand seems to give a slight underestimation of the coherence. The wavenumber at which the measured coherence dropped to the level of 0.5 is 0.027 rad/m for the 2-beam and 0.051 rad/m for the 4-beam. These wavenumbers have been defined as the smallest detectable eddy size (Schlipf et al., 2018) and can be interpreted as the size of the eddy that is captured with an accuracy of 50%. They are approximately 219.4 m ($4.2D$) for the 2-beam and 122.5 m ($2.6D$) for the 4-beams

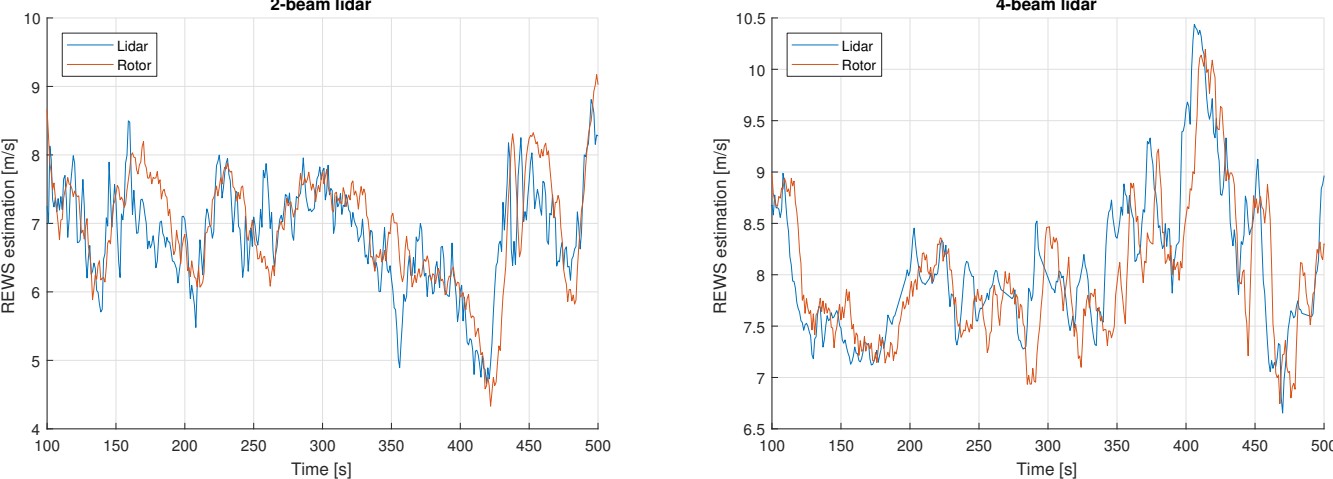

**Figure 8.** Time series example of lidar and turbine estimates of REWS. A high degree of similarity between the signals can be seen. Also the preview ability of the lidar system is evident.

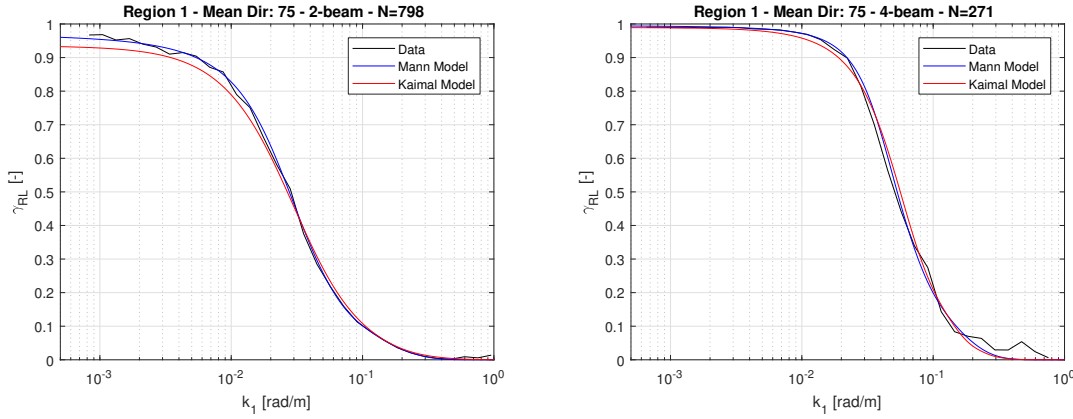

**Figure 9.** Squared coherence between the REWS estimation of the lidar and the turbine for region 1. The two models are also included in the plot.

system, where the number in the brackets are normalized by the rotor diameter. Thus, by adding two additional focal points to a 2-beam nacelle lidar system the smallest detectable eddy size can be reduced by 44%.

The results for region 2 are presented in fig. 10. Here very similar observations can be made. The coherence for the 2-beam lidar drops at lowers wavenumbers compared to the 4-beam. The wavenumbers at $\gamma_{\mathrm{RL}} = 0.5$ are $0.032\,\mathrm{rad/m}$ and $0.056\,\mathrm{rad/m}$ and the smallest detectable eddy sizes are $198.8\,\mathrm{m}$ ($3.8D$) and $111.4\,\mathrm{m}$ ($2.1D$), respectively. This demonstrates once more a reductions of 44% in the smallest detectable eddy size. Comparing these numbers to the results of region 1 shows that flow having larger length scale parameter is beneficial for lidar systems as the coherence drops at higher wavenumbers.

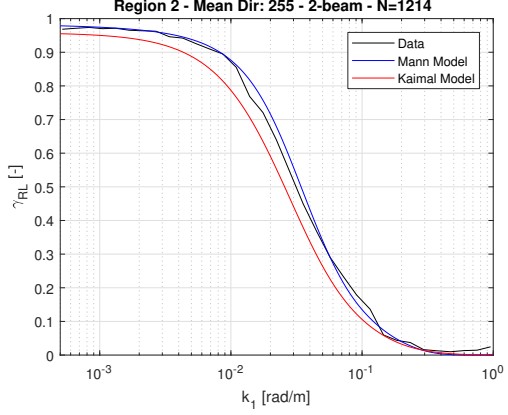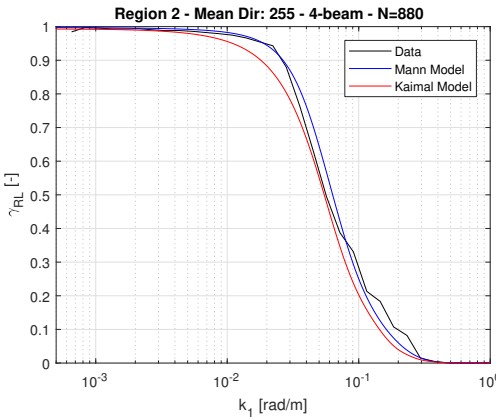

**Figure 10.** Squared coherence between the REWS estimation of the lidar and the turbine for region 2. The two models are also included in the plot.

Equivalently to region 1, the Mann turbulence model fits very well to the measured data. There are however some slight deviations for both lidars in the region of 0.01 to 0.1 rad/m, which could be caused by modeling inaccuracies of the assumed turbine model or general measurement noise in the turbine or lidar data. When comparing the experimental data to the Kaimal model, a larger mismatch is observed compared to region 1. These deviations could stem from the lack of the Kaimal model
5 to represent 3D turbulent structures. It is a 1D model, which has been extended to represent 3D turbulence by applying an empirical exponential lateral coherence model with completely independent velocity components, while the Mann model defines a full 3D tensor model. Thus, the Mann model is able to represent better the three-dimensional structure of turbulence and thus give more realistic coherences.

To quantify the error between measured and model-derived coherence, we use the root-mean-squared error (RMSE) calculated from the data presented in fig. 9 and 10. The RMSE is defined as

$$\text{RMSE} = \sqrt{\frac{1}{N}\sum_{i=1}^{N}(\gamma_{\text{RL,Data,i}} - \gamma_{\text{RL,Model,i}})^2},$$

where $\gamma_{\text{RL,Data,i}}$ refers to the measured coherence and $\gamma_{\text{RL,Model,i}}$ is the model-derived coherence and $N$ is the number of data
10 points. The model-derived coherence has been linearly interpolated at the available wavenumbers from the measurements. The RMSE is summarized in tab. 3.

**Table 3.** Comparison of the RMSE between measured and model-derived coherence for the 2- and 4-beam lidar in region 1 and 2.

| | 2-beam | | 4-beam | |
|---|---|---|---|---|
| | Region 1 | Region 2 | Region 1 | Region 2 |
| Mann Model $[10^{-2}]$ | 0.955 | 1.900 | 1.994 | 2.662 |
| Kaimal Model $[10^{-2}]$ | 2.194 | 4.842 | 2.395 | 3.850 |

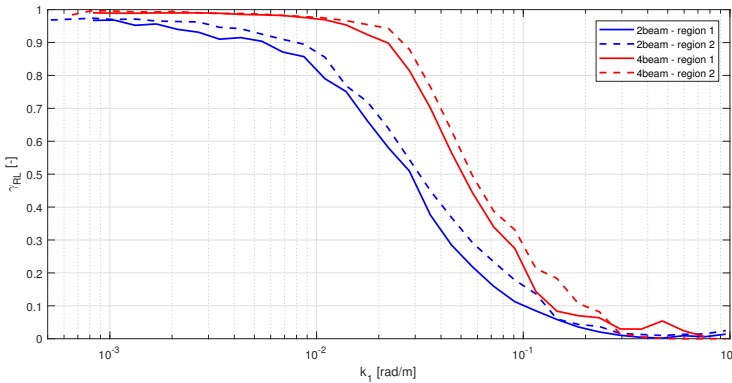

**Figure 11.** Comparison of the measured coherence of the 2- and 4-beam lidar for regions 1 and 2.

It can be seen that the errors between measured data and the model based on the Mann turbulence model are consistently lower than those of the model based on the Kaimal turbulence model. For the 2-beam lidar the error are approximately twice as high, while the difference is slightly less for the 4-beam lidar. Also, in region 2 the model based on the Kaimal turbulence model performs worse, which also can be seen when comparing fig. 9 and 10.

Next, to better compare the measured coherence of the 2- and 4-beam lidar systems for region 1 and 2, all measured coherences have been plotted in one figure. It can be seen that the coherence of the 4-beam system drops at larger wavenumbers than the 2-beam lidar due to probing the incoming wind at 2 additional positions. Difference between region 1 and 2 can also be observed. For both lidars, the coherence measured in region 2 is higher than region 1. This can be explained by the larger turbulence length scale, which implies that there are more large-scale fluctuations in the flow. These large-scale fluctuations can be better resolved by both lidar systems.

Further, it is observed that even without the inclusion of the evolution of turbulence the model is able to predict the coherence very accurately. This implies that effect of disregarding turbulence evolution can be neglected. For bigger turbines, where larger focus distances are required, turbulence evolution might become more significant.

### 4.3 Time delay analysis

Next, the delay between lidar and turbine estimations of REWS are analyzed. The delay stems from the perpendicular distance between the rotor plane and the measurement plane $\Delta x$. It depends on the advection speed

$$\Delta t = \frac{\Delta x}{U_{\mathrm{adv}}} - \frac{t_{\mathrm{Scan}}}{2}, \tag{25}$$

where $t_{\mathrm{Scan}}$ is the time to perform one full scan, which is 1 s for both lidars and $\Delta x = d_f \cos\alpha - d_{\mathrm{Nac}}$, where $d_{\mathrm{Nac}} \approx 1\,\mathrm{m}$ is the distance between lidar mounting position and rotor. $U_{\mathrm{adv}}$ is the advection speed of the turbulent fluctuations, which is estimated by the 10 minute average of the lidar estimated REWS: $U_{\mathrm{adv}} \approx \overline{\hat{v}_{\mathrm{eff,L}}}$.

Since the experiment was performed with very good time synchronization (with a maximum time delay of a few $\mu$s), it is also possible to calculate the delay between the two signals and compare it with expected delays based on the advection speed. The

delay between the two signal has been calculated using the information theoretical delay estimator presented in Moddemeijer (1988). This method is based on splitting the two input signal into two parts: the past and the future and calculating the mutual information of two signals between the past and the future signal. The input signals are split in the middle of the time series. By shifting the signals relative to each other, the time delay which minimizes the mutual information is found. We have found that this method performs better than other delay estimators, namely the maximum index of the cross-correlation and the slope of the cross-spectrum. Due to the sampling rate of 1 Hz calculated delays are discretized in steps of 1 s.

The result can be seen in fig. 12. For the 2-beam lidar shorter delays are expected due to the smaller focus distance and larger half-cone opening angle. Still, the results for both lidars show great overlap between the measured delay and the delay expected from advection speed and the lidar geometry. The match of the measured and expected delays is worse for the 2-beam lidar compared to the 4-beam lidar. An overestimation of the delay for high wind speeds can be observed. However, the delay for high wind speeds is small and the delay estimator is affected by the low sampling rate of 1 Hz, which results in a delay resolution of 1 s. For low wind speed a underestimation can be observed in the 2-beam lidar data, which can also be identified to some extend in the 4-beam lidar data. We speculate that this can be explained by faster wind direction changes at low wind speeds which lead to a turbine misalignment.

In general, towards high wind speeds the available preview time provided by the lidar becomes smaller. The required preview time for the filtering is also shown for the two lidar setups. Low-pass filtering the lidar system is crucial to reject high-frequent fluctuations that are sensed by the lidar but not experienced by the rotor and if not filtered would cause detrimental pitch actuation. In this study a first-order Butterworth filter is used following the approach presented in Schlipf (2015), though different filters have been proposed, e.g. a Wiener filter (Simley and Pao, 2013a). The delay of the filter is nonlinear but can be approximated by the delay at a certain frequency of interest $\omega_{\mathrm{delay}} = 2\pi f_{\mathrm{delay}}$ (Schlipf, 2015):

$$t_{\mathrm{filt}} = \frac{\arctan\left(\frac{\omega_{\mathrm{delay}}}{\omega_{\mathrm{cutoff}}}\right)}{\omega_{\mathrm{delay}}}, \tag{26}$$

where $f_{\mathrm{delay}} = 0.0425\,\mathrm{Hz}$ was chosen as the frequency where the maximum of the rotor speed spectrum was observed. The cutoff frequency was determined from the coherence analysis presented previously at the point where $\gamma_{\mathrm{RL}} = 0.5$. The average over region 1 and 2 was computed for both lidar systems and $\omega_{\mathrm{cutoff}} = U_{\mathrm{adv}}k|_{\gamma_{\mathrm{RL}}=0.5}$. This implies that the cutoff frequency changes with advection speed and an adaptive filter is required. For the 2-beam lidar, it can be seen that there is sufficient time to perform the filtering at low wind speeds, but at high wind speed there is not enough preview time for the filtering. This results implies that a larger focus distance or smaller opening angle should be chosen for the 2-beam lidar in order to be able to perform the low-pass filtering. For the 4-beam lidar, it can be seen that the expected preview time provided by the lidar system is sufficient for the low-pass filtering.

## 5   Conclusions

In this study we presented a model of the coherence between REWS estimated from turbine and lidar measurements. The underlying model of the 3D turbulent field is the Mann spectral tensor and allows the direct calculation of auto- and cross-

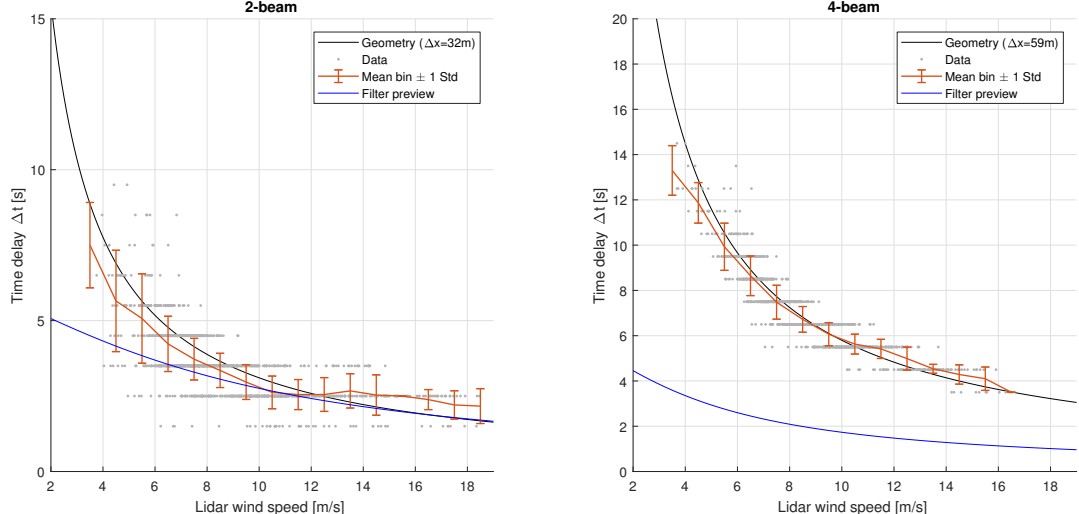

**Figure 12.** Delay time analysis results for the 2-beam and 4-beam lidar. Delays and advection speeds are calculated based on 10 minute averages. The required filter time of a first-order Butterworth filter is also shown.

spectra of REWS estimations for lidar and turbine. It is compared to field data obtained from two continuous-wave lidar systems mounted on top of the nacelle of a wind turbine. To retrieve the turbulence model parameters, measured spectra from a sonic anemometer have been fitted to the spectral tensor. The comparisons of squared coherence show that the presented model fits the field data better than previously used models, which are based on the Kaimal model defined in IEC standard.

Thus, this study gives confidence that the proposed model can accurately represent the important lidar properties and it can be used to optimize the lidar focus point positions to maximize the coherence between lidar and turbine. A common parameter used in the lidar optimization is the wavenumber where the coherence drops to a value of 0.5 (Dunne et al., 2014), which can be calculated precisely by the model.

We have found that larger turbulence length scales led to higher coherences between REWS estimated of turbine and lidar

compared to inflow turbulence of smaller length scale. It was also shown that the smallest detectable eddy size can by reduced by almost 50% when using the 4-beam compared to the 2-beam system. Further, the advection speed by which the turbulent structures are transported from measurement to rotor plane can be estimated from 10 minute averages of REWS from lidar measurements. This is important information for the correct timing of the measured fluctuations of the lidar systems. In case of the 4-beam lidar there is enough preview provided by the lidar to perform the necessary low-pass filtering, while the 2-beam

lidar lacks preview time for filtering for high wind speeds.

Since some of the physical mechanisms have not been modelled, future work includes additions to both the lidar and turbulence modelling. First of all, the evolution of turbulence as it travels from measurement to rotor plane has been neglected. An amendment of turbulence evolution to the Mann model has been proposed in de Maré and Mann (2016). The evolution will have most influence on the small-scale fluctuations (Bossanyi, 2013) and including the effect will reduce the coherence.

Hence, the model presented here can be considered an idealized case. On the other hand, only small differences were observed between data and model implying that the evolution effect is small. For larger turbine, which require larger focus distances, this effect could be more severe. Secondly, the stability of the atmosphere was not considered, i.e. a neutral stratification was assumed. Extensions to the Mann model have been proposed to include effect of the atmospheric stability, e.g. Segalini and Arnqvist (2015) or Chougule et al. (2018). It should be noted that the discrete scanning of the lidar system and possible blade blockage effects have not been integrated into the model. Also, environmental conditions like the aerosol concentration, fog or precipitation have been disregarded.

The presented model in the current form can be applied to nacelle-mounted cw lidar and by modifying the spatial averaging of the lidar it can be extended to nacelle-mounted pulsed lidars as well. To cover spinner-mounted lidar systems the rotational sampling effect of the lidar as it rotates with the rotor needs to be modelled.

## Appendix A: Calculation of the power coefficient surface

For the calculation of the $C_p(\beta, \lambda)$ surface an aerodynamic model of the Vestas V52 turbine was used. Aero-elastic simulations in HAWC2 over a domain of several pitch angles and TSR have been performed. A homogeneous, constant wind speed of 8 m/s was used and constant pitch angles and rotational speeds of the rotor were set during the simulation. Stiff tower and blades were used to avoid dynamic effects and calculate quasi-steady state $C_p$ values. The resulting surface can be found in fig. A1.

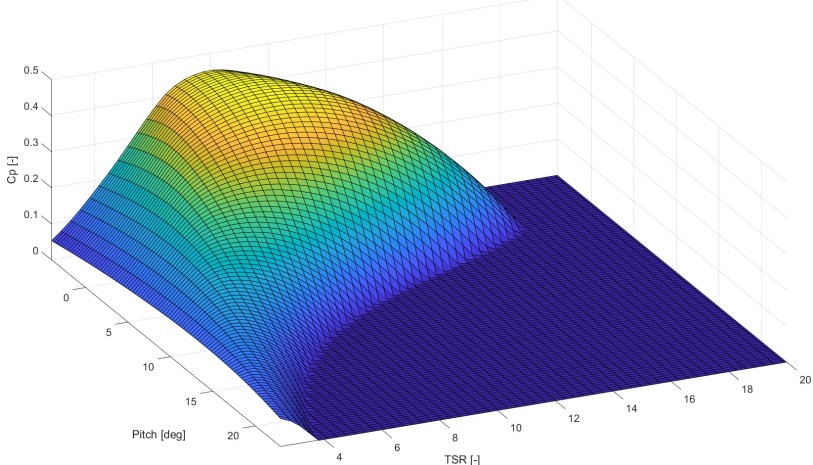

**Figure A1.** Calculated $C_p(\beta, \lambda)$ surface for the Vestas V52 turbine. For illustrative purposes negative $C_p$ values have been replaced with zero.

## Appendix B: Induction Correction

In this appendix an example of the flow field around a rotor operating at the aerodynamic optimum according to the model of Conway (1995) is shown. The focus positions of the 4-beam lidar are indicated in red.

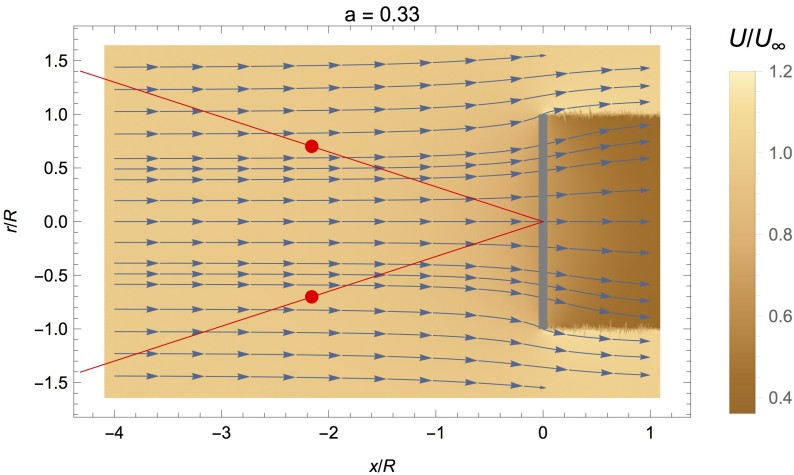

**Figure B1.** Example of the flow speed reduction and diversion around the rotor of a wind turbine. The red lines indicate the laser beam and the red dots show the focus points.

## Appendix C: Comparison between average fitted Mann model parameter and average spectra

In section 3.2 the analysis was split into two distinct region, of which region 1 was disturbed by buildings and vegetation, while region had a more undisturbed inflow over field and the fjord. Previously, the average spectra were fitted to the Mann model for sectors of 30° width. The fitted parameter were then averaged per region to obtain a representative set of parameters for each region. Here the averaged parameter are compared to the average spectra for region 1 and 2. In the process each individual $u$-, $v$- and $w$- spectrum and the $uw$ co-spectrum has been normalized by the 10 minute mean wind speed squared and the average over all spectra for each region was taken. The result can be seen for region 1 in the left graph of fig. C1 and for region 2 in the right graph of fig. C1. It can be seen that in both cases the model is describing the $u$-, $v$- and $w$- spectrum well. The $uw$ co-spectrum is underestimated by the model, which was also found in the fits done in Mann (1994).

*Author contributions.* Dominique P Held performed the research work and prepared the manuscript. Jakob Mann conceived the research plan, supervised the research work and the manuscript preparation.

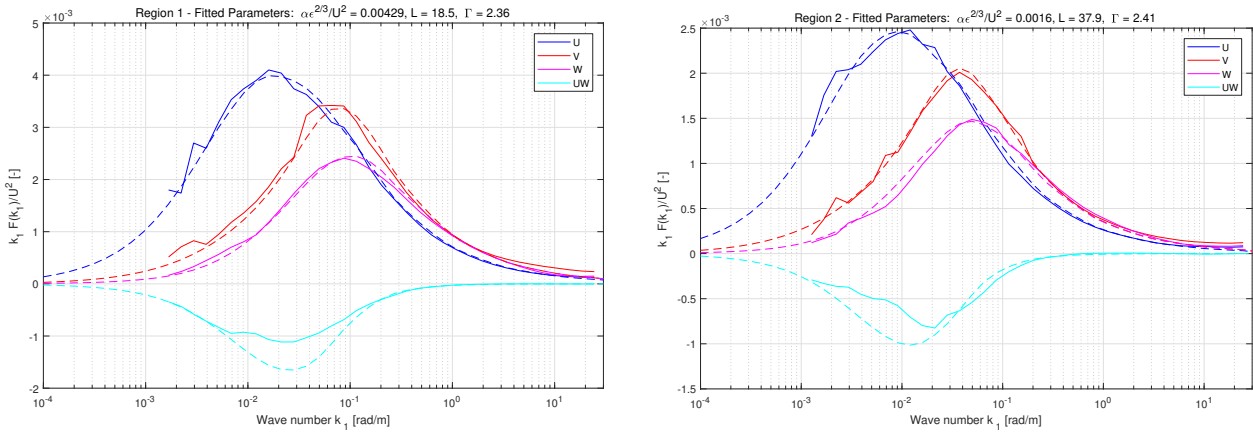

**Figure C1.** Average spectra and the corresponding Mann model fits from tab. 2 for region 1 (*left*) and region 2 (*right*).

*Competing interests.* The work of Dominique P Held was partly funded by Windar Photonics A/S in form of an industrial PhD stipend (project number: 5016-00182).

*Acknowledgements.* This study was supported by Innovationsfonden Danmark in form of an industrial PhD stipend (project number: 5016-00182). The authors also want to acknowledge the research project *Cost-efficient lidar for pitch control* funded by Energiteknologiske Udviklings- og Demonstrationsprogram and specifically Hector Villanueva for providing the turbine data in form of a database, Qi Hu for designing and constructing the 4-beam lidar, Ebba Dellwik for the coordination within the project and the technicians at DTU Wind Energy for conducting the experiments on the Vestas V52 turbine at the Risø test site. The authors also want to thank Pedro Santos and Ebba Dellwik for their comments and suggestions to the manuscript.

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
