# Peer review of "Lidar Estimation of Rotor-Effective Wind Speed - An Experimental Comparison"

_Wind Energy Science, 2018_

## Referee Comment (RC1) · Anonymous Referee #1 · 30 Jan 2019

General Comments:

This paper presents a spectral model for estimating the coherence between nacelle-based continuous wave lidar measurements and the rotor effective wind speed using the Mann turbulence model. The expected coherence from the model is then compared with the measured coherence for two types of nacelle lidars from Windar Photonics mounted on a Vestas V52 turbine. The measurement coherence is very relevant to the design of lidar-assisted control of wind turbines, making the paper a valuable contribution. In general, the modeled coherence matches the measured coherence very well. Previous papers have used the Kaimal model for such comparisons, but the authors

show that the Mann model fits the field data better. The manuscript also presents one of the most thorough comparisons between modeled coherence and field-measured coherence in the literature. In addition, the present work includes the influence of different turbulence length scales on the measurement coherence, showing that the model and field measurements change in similar ways to the turbulence parameters. The paper is very well written and overall explains the methodology well. There are places where I believe more explanation of the methods should be included, however, and I believe some of the results could be presented more clearly. These and other suggestions for improvement are included below.

Specific Comments:

Section 1: Another relevant publication, which compares the theoretical coherence bandwidth using the Kaimal model for different lidar scan patterns, including 2-beam and 4-beam lidars, is:

Simley, Eric and Fürst, Holger and Haizmann, Florian and Schlipf, David, Optimizing Lidars for Wind Turbine Control Applications - Results from the IEA Wind Task 32 Workshop, Remote Sensing. 2018

Eqs. 16, 18, 19: Some derivation details (like Eqs. 6-8) or references to sources where these equations are derived would be appreciated.

Eqs. 18, 19: Please clarify whether you are modeling the sequential scanning, or assuming all beams are measured simultaneously.

Pg. 7, ln. 12: For the yaw misalignment correction, can you explain if you are trying to estimate the total horizontal wind speed, or the component perpendicular to the rotor? Additionally, comment on differences between measurements with the corrected velocities and the spectral model. For example, with yaw misalignment the measured wind will travel toward the rotor at an angle and reach the rotor at a different position than the model assumes. This could cause some differences between the measurements

[Figure]

and model that the correction doesn't account for.

Table 2: Do you notice differences in length scales and other parameters if you bin by stability in addition to by sector, and would this be worth considering in the analysis?

Figs. 9 and 10: A very important finding of this study is that even without including wind evolution, the measured coherence is very close to the modeled coherence, suggesting that wind evolution is not one of the main sources of error when estimating the rotor effective wind speed with a lidar. I think this is a key result and should be highlighted more.

Figs. 9 and 10: It would be easier to interpret the coherence for the 2-beam vs. 4 beam and region 1 vs. region 2 if the coherence curves for different cases were plotted in the same plot. At least a plot comparing the measured coherence curves for the four cases would make it easier to compare.

Pg. 15, ln. 4: "Comparing these numbers to the results of region 1 shows that flow having larger length scale parameters is beneficial for lidar systems..." In addition to the length scales being larger for region 2, the viscous dissipation of turbulent kinetic energy is lower. Could this also lead to improved coherence?

Pg. 15, ln. 6: "There are however some slight deviations for both lidars in the region of 0.01 to 0.1 rad/m." What are some possible reasons for this mismatch?

Pg. 15, ln. 7: "When comparing the experimental data to the Kaimal model, a larger mismatch is observed compared to region 1." It appears that the coherence for the Kaimal model does not change much between Figs. 9 and 10, and that the Mann model changes similar to the field measurements. Can you comment on this?

Eq. 22: Would you expect the induction zone to slow down the advection speed? Does this appear in the field-measured time delay?

Pg. 15, ln. 20: "the information theoretical delay estimator" This sounds like a great way to estimate the time delay. Could you briefly clarify how the two input signals are

split into past and future? Are you comparing the past part of one signal to the future part of the other?

Fig. 11: Because there is so much scatter in the field data, to understand the trend of the field time delays, the average time delay binned by wind speed would be valuable to show in the plots.

Pg. 16, ln. 14: Choosing 0.433 Hz for the delay frequency does not seem like the best choice, especially because it is much higher than the cutoff frequency. It would be better to base the delay frequency on the frequency where the wind disturbance impacts the signal of interest (like rotor speed) the most. This could be found with a linear model of the closed-loop turbine. Doing this would provide a more realistic value for the preview time needed (but I imagine would still be within the available preview time you have observed). See for example (Schlipf, 2015), where the delay frequency is chosen as 0.1 Hz, which is the frequency where peak of the rotor speed spectrum is located.

Technical Corrections:

Pg. 2, ln. 12: "exponential decay model" -> "exponential decay coherence model"

Pg. 2, ln. 19: "...if both quantities want to be measured" would sound better as (for example) "...if measurements of both quantities are wanted"

Pg. 3, ln. 15: "where a reduction in the blade and tower DELs..." -> "where the blade and tower DELs..."

Pg. 5, ln. 26: "non-monotony" -> "non-monotonicity"?

Eq. 16: Should "k \cdot x" be "k \cdot n"?

Pg. 7, ln. 11: Consider using a different symbol for turbine misalignment since phi is already used for the weighting function.

---

## Referee Comment (RC2) · Anonymous Referee #2 · 4 Feb 2019

The paper presents an experimental comparison of the rotor-effective wind speed (REWS) from lidar and turbine data. Overall, the paper is very well-written and includes very important findings: the correlation of lidar preview to the turbine reaction can be better modeled by the Mann turbulence model compared to Kaimal model used in previous work, even if wind evolution is not included. This is very relevant for lidar-assisted control application. The analysis is done based on the large data set (should be the largest published so far) and the paper includes interesting details, e.g. a fully analytic model of the correlation. Although the quality of the paper is already very high, the issues mentioned below might be helpful to further improve the paper.

[Figure]

Time delay:

The calculation of the time delay via the "information theoretical delay estimator" is an interesting approach. However, the information in Figure 11 is hard to read. Couldn't you use mean and standard deviation for each discrete time delay? Or wind speed bin? Further, the calculation of the filter preview can be improved. First, the frequency at 1P is quite high since the frequencies you are still able to measure with a good coherence are much lower. Also, the coherence level of 0.5 might be not a good cut-off-frequency, depending on the lidar spectrum. E.g. rotor and lidar estimate of the REWS can have a coherence of 1 and still have different spectra (if the transfer function is linear). In this case, using the coherence level of 0.5 would lead to no filtering, but filtering would be necessary depending on the transfer function. The transfer function at -3dB (for first order linear filter, approx. 0.7) should be better suited, especially for lidars with little averaging effect as the ones presented in this work. The 0.5 coherence level should be only close to the -3dB, if S_LL is quite similar to S_RR. Thus, would be good to better motivate the coherence level of 0.5 or use the transfer function at -3dB.

Details to improve understanding:

- Eq. 3 and 4: What is \delta, k'?

- Eq. 9: Some intermediate steps how to get there might be helpful. Maybe in the appendix? How do they relate to the equations from Mirzaei and Mann (2016)?

- Eq. 20 and 21: It is also not really clear, how the correction is applied to the real data, since only 1 Hz data are collected. Is this algorithm done on the lidar system or in post processing? It is also not clear, where this correction comes from. Line-of-sight wind speeds are often used in a wind field reconstruction algorithm which directly provides derived signals such as the REWS. And maybe I am wrong, but the correction seems to be the same than reconstructing the average horizontal wind speed. For small misalignment angles it might be not very important. For larger angles however, it is more the average longitudinal wind having an "effect" on the rotor. Thus it is not

clear, why this correction is necessary. But maybe I missed something. Thus, some explication might be helpful.

Organization:

The paper is mostly well organized. Only Section 4 might be separated into subsections and Section 2 might be better organized. The part before the current 2.1 could be included in a subsection "Overview Coherence Model". Currently, 2.1 is including the model of the rotor spectrum, 2.2. how you get the REWS estimate from turbine data. Then 2.3 combines model of the lidar spectrum, cross spectrum and how you get the REWS estimate from lidar data. Thus, the subsections seem to be not on the same level. For the understanding, it might be better to first describe the model and its component (2.2 $S\_RR$, 2.3 $S\_LL$, 2.4 $S\_RL$) and then the model implementation and validation against simulation (2.5) and then the modification for field testing (turbine measurements).

Minor issues (please ignore them, if too picky)

Eq. 20: Shouldn't \beta be \beta_i?

p20, l1: Up to this point, it is not mentioned that both lidar systems provides 1 Hz data. Table one might lead to 2 and 4 Hz. So it might be not clear at this point why >90% is equal to 540 measurements).

Figure 1: \theta_{FF} and \theta_{FB}. In text on page 1: FF and FB are not in mathmode.

Captions of Fig. 9 and 10 don't end with a period, others do.

Figure 3,6,7: unit in labels (partly) missing.

---

## Referee Comment (RC3) · Anonymous Referee #3 · 5 Feb 2019

The main purpose of this paper is to compare the Mann coherence model to the Kaimal model when considering lidar measurement of the rotor-effective wind speed. The work paper is generally well-written and the measurement campaign produced an impressive data set. However, I believe that a more quantitative approach should be used in comparing the two models.

My major suggestions are the following:

1. When comparing the Mann model to the Kaimal model, a more statistical approach would be preferred. For instance, on page 13 line 15, it is stated that "... the measured data agrees very well with the Mann turbulence model coherence. The Kaimal

model on the other hand seems to give a slight underestimation of coherence". Can you quantify this agreement/underestimation numerically? Perhaps as a function of frequency?

2. Could you provide concisely the benefits of the Mann model over the Kaimal model, and potential drawbacks of the Mann model if applicable? Computation time is mentioned on page 2, lines 13-14, but not elaborated on; and Mann model fitting is mentioned in Appendix C, but it is unclear to me how much more complex the Mann model (3D) is than the Kaimal model (1D) (page 15, line 9).

3. In the conclusion, it is stated that blade passage effects were not included. In the experimental data, how was the effect of blade passage removed?

I also consider the following points to be important:

a. A fuller description of lidar operation would be preferred, for readers who are not highly familiar with lidar operation. There are terms used that may not be very meaningful to readers, such as Rayleigh length, Doppler spectrum, probe volume, focus distance, and perhaps even wave number.

b. Is (3) an alternative definition of the spectral tensor? The definition from Mann (1994) does not appear in the manuscript.

c. I don't understand the i,j indices in (16). Further, the meaning of ". . . summation of repeated indices is implied" (page 6, line 14) is unclear to me. It appears also that in (18) and (19), the i,j indices are replaced with k,l. Could you be explicit about what each of these indices represent (especially since k has been assigned to the wavenumber previously)?

d. Labeling on plots needs some improvement. Font sizes are small and labels are unclear. For instance, Figure 6 has "WSP" on the x-axis, but this is not defined.

e. There is inconsistent notation, particularly with Greek lettering. $\beta$ is used for both pitch angle and azimuth angle, e.g., page 5, line 20 and table 1. $\beta$ and $\theta$ are both

used for blade pitch angle, e.g., page 1, line 20 and page 5, line 20. $\varphi$ is used for both the lidar weighting function (eq. (15)) and the average turbine misalignment (page 7 line 11). alpha is used for both the half-cone opening angle (page 6, line 21) and the Kolmogorov constant (page 11 line 1).

f. It would be nice if some of the key parameters are labeled in the left panel of Fig. 4. For instance, the half-cone angle and the focus distance. The distance d_{Nac} could be labeled in the right panel of Fig. 4, especially that this is not really defined until page 15 (long after table 1 where it first appears). Delta x, which also appears in Table 1, was not completely clear to me until page 15 when it is finally defined.

g. Figures 9 and 10 could be labeled more clearly. Perhaps the "Data" line should be labeled "turbine estimates" (or something similar)? Then it may be clearer that each curve represents the coherence between the lidar data and each of the labeled items.

The following are smaller corrections and suggestions for improvement:

I. There is inconsistent notation for lidar focus distance (d_f vs L_f). See page 6, line 8 and page 7, line 7.

II. What is the difference between the induction factor and the axial induction factor on page 7? What is the relationship between them?

III. The term 'filter' appears to be used to refer to both a digital filter (in referring to a Butterworth Filter, page 16) and something far more general on page 12. This use (particularly the more general use, on page 12) is confusing in a scientific journal.

IV. Readers may find the use of 'region 1, region 2' (page 13 lines 11-12) confusing if coming from a wind energy background, where modes of turbine operation (below cut-in, below rated operation) are also referred to as 'regions'.

V. There are some typos and grammatical errors throughout which should be corrected.

---

## Author Comment (AC1) · 12 Apr 2019

Please find the responses to the review comments in the supplement. There is also a manuscript version where the changes made are indicated.

Please also note the supplement to this comment: https://www.wind-energ-sci-discuss.net/wes-2018-72/wes-2018-72-AC1-supplement.zip

───────────────────────

---

## Author Response (AR1)

**Author Response to Review Comment #1**

Dear Reviewer,

Thank you for reviewing the manuscript. Your comments were very helpful and improved the quality of the manuscript. The author responses can be found below each reviewer comment.

| | |
|---|---|
| RC 1.1 | Section 1: Another relevant publication, which compares the theoretical coherence bandwidth using the Kaimal model for different lidar scan patterns, including 2-beam and 4-beam lidars, is: Simley, Eric and Fürst, Holger and Haizmann, Florian and Schlipf, David, Optimizing Lidars for Wind Turbine Control Applications - Results from the IEA Wind Task 32 Workshop, Remote Sensing. 2018 |

AC  This reference has been added and a description of the main finding has been added to the introduction section.

| | |
|---|---|
| RC 1.2 | Eqs. 16, 18, 19: Some derivation details (like Eqs. 6-8) or references to sources where these equations are derived would be appreciated. |

AC  References to the a paper where details of the derivation of the formulas can be found have been added below the formulas. The reference is also found at the beginning of the section.

| | |
|---|---|
| RC 1.3 | Eqs. 18, 19: Please clarify whether you are modeling the sequential scanning, or assuming all beams are measured simultaneously. |

AC  Here simultaneous measurements are assumed. This has been clarified in the text.

| | |
|---|---|
| RC 1.4 | Pg. 7, ln. 12: For the yaw misalignment correction, can you explain if you are trying to estimate the total horizontal wind speed, or the component perpendicular to the rotor? Additionally, comment on differences between measurements with the corrected velocities and the spectral model. For example, with yaw misalignment the measured wind will travel toward the rotor at an angle and reach the rotor at a different position than the model assumes. This could cause some differences between the measurements and model that the correction doesn't account for. |

AC  Here we trying to estimate the component perpendicular to the rotor. A clarification has been added below eq. 21. Also differences to the model have been pointed out. But in case of small yaw alignment the expected differences are assumed to be small.

| | |
|---|---|
| RC 1.5 | Table 2: Do you notice differences in length scales and other parameters if you bin by stability in addition to by sector, and would this be worth considering in the analysis? |

AC   We have considered binning by stability as the Mann turbulence model is a representation of turbulence in the neutral atmosphere. However, we saw no deviations of the measured point spectra to the fitted Mann turbulence model for different stability classes, see appendix C. Thus, we did not divide the data into stability classes.

RC 1.6   Figs. 9 and 10: A very important finding of this study is that even without including wind evolution, the measured coherence is very close to the modeled coherence, suggesting that wind evolution is not one of the main sources of error when estimating the rotor effective wind speed with a lidar. I think this is a key result and should be highlighted more.

AC   We agree to this statement and have added a paragraph in the result section.

RC 1.7   Figs. 9 and 10: It would be easier to interpret the coherence for the 2-beam vs. 4 beam and region 1 vs. region 2 if the coherence curves for different cases were plotted in the same plot. At least a plot comparing the measured coherence curves for the four cases would make it easier to compare.

AC   We have added an additional figure for the measured coherences, where the measured coherence for the 2- and 4-beam lidar systems is shown for both region 1 and 2. A paragraph summarizing the results has also been added.

RC 1.8   Pg. 15, ln. 4: "Comparing these numbers to the results of region 1 shows that flow having larger length scale parameters is beneficial for lidar systems. . ." In addition to the length scales being larger for region 2, the viscous dissipation of turbulent kinetic energy is lower. Could this also lead to improved coherence?

AC   According to the mode, the viscous dissipation of turbulent kinetic energy ($\varepsilon$) does not have an influence on the coherence since the spectra depend linearly on $\varepsilon$ and thus cancel each other out. This is also observed from the measurements, where increased $\varepsilon$ in region 2 do not lead to biases in the agreement with the model.

RC 1.9   Pg. 15, ln. 6: "There are however some slight deviations for both lidars in the region of 0.01 to 0.1 rad/m." What are some possible reasons for this mismatch?

AC   Possible reasons for the mismatch can be measurement noise in the lidar or turbine data and modeling inaccuracies when calculating the REWS from turbine data. These points have been added to the section.

RC 1.10   Pg. 15, ln. 7: "When comparing the experimental data to the Kaimal model, a larger mismatch is observed compared to region 1." It appears that the coherence for the Kaimal model does not change much between Figs. 9 and 10, and that the Mann model changes similar to the field measurements. Can you comment on this?

AC   The coherence changes are bigger for the Mann turbulence model because changes in the three-dimensional structure of turbulence are better represented by the Mann turbulence model. The Kaimal model is less flexible at representing changes in the three-dimensional structure of the turbulence and thus smaller changes are seen between the two regions for this model.

RC 1.11   Eq. 22: Would you expect the induction zone to slow down the advection speed? Does this appear in the field-measured time delay?

AC   The expected time delay has been calculated based on the estimated lidar REWS, which has been corrected for the induction effect based on turbine measurements and the position of the focus points. So in a simple way we have included the slow-down due to induction. We have not found evidence of any biases introduced by the turbine's induction effect.

RC 1.12   Pg. 15, ln. 20: "the information theoretical delay estimator" This sounds like a great way to estimate the time delay. Could you briefly clarify how the two input signals are split into past and future? Are you comparing the past part of one signal to the future part of the other?

AC   The signal is split in the middle. Then both past values of the signal are compared to the both future values. This has been clarified in the manuscript. More details can be found in the reference.

RC 1.13   Fig. 11: Because there is so much scatter in the field data, to understand the trend of the field time delays, the average time delay binned by wind speed would be valuable to show in the plots.

AC   We have added binned mean values and +-1 standard deviations.

RC 1.14   Pg. 16, ln. 14: Choosing 0.433 Hz for the delay frequency does not seem like the best choice, especially because it is much higher than the cutoff frequency. It would be better to base the delay frequency on the frequency where the wind disturbance impacts the signal of interest (like rotor speed) the most. This could be found with a linear model of the closed-loop turbine. Doing this would provide a more realistic value for the preview time needed (but I imagine would still be within the available preview time you have observed). See for example (Schlipf, 2015), where the delay frequency is chosen as 0.1 Hz, which is the frequency where peak of the rotor speed spectrum is located.

AC   Yes, we agree to the statement. We have plotted the average spectra of the rotor speed for above-rated wind speeds below. The maximum of this spectrum is at 0.0425 Hz. We have used this as the delay frequency and redone the time delay analysis. The 2-beam lidar is lacking preview time at high wind speeds, which implies that a larger distance between measurement and rotor plane should be chosen.

[Figure]

RC 1.15   Pg. 2, ln. 12: "exponential decay model" -> "exponential decay coherence model"

AC   This has been changed in the manuscript.

RC 1.16   Pg. 2, ln. 19: ". . .if both quantities want to be measured" would sound better as (for example) ". . .if measurements of both quantities are wanted"

AC   This has been changed in the manuscript.

RC 1.17   Pg. 3, ln. 15: "where a reduction in the blade and tower DELs. . ." -> "where the blade and tower DELs. . ."

AC   This has been changed in the manuscript.

RC 1.18   Pg. 5, ln. 26: "non-monotony" -> "non-monotonicity"?

AC   This has been changed in the manuscript.

RC 1.19   Eq. 16: Should "k \cdot x" be "k \cdot n"?

AC   This has been changed in the manuscript.

RC 1.20   Pg. 7, ln. 11: Consider using a different symbol for turbine misalignment since phi is already used for the weighting function.

AC   "\varphi" has been replaced by "\phi".

**Author Response to Review Comment #2**

Dear Reviewer,

Thank you for reviewing the manuscript. Your comments were very helpful and improved the quality of the manuscript. The author responses can be found below each reviewer comment.

| | |
|---|---|
| RC 2.1 | Time delay: The calculation of the time delay via the "information theoretical delay estimator" is an interesting approach. However, the information in Figure 11 is hard to read. Couldn't you use mean and standard deviation for each discrete time delay? Or wind speed bin? Further, the calculation of the filter preview can be improved. First, the frequency at 1P is quite high since the frequencies you are still able to measure with a good coherence are much lower. Also, the coherence level of 0.5 might be not a good cut-off-frequency, depending on the lidar spectrum. E.g. rotor and lidar estimate of the REWS can have a coherence of 1 and still have different spectra (if the transfer function is linear). In this case, using the coherence level of 0.5 would lead to no filtering, but filtering would be necessary depending on the transfer function. The transfer function at -3dB (for first order linear filter, approx. 0.7) should be better suited, especially for lidars with little averaging effect as the ones presented in this work. The 0.5 coherence level should be only close to the -3dB, if S_LL is quite similar to S_RR. Thus, would be good to better motivate the coherence level of 0.5 or use the transfer function at -3dB. |

AC    Yes, we agree that the readability of fig. 11 can be improved. We have added the bin mean and +-1 standard deviation to the plot.
For a discussion on the 1P frequency please see review comment 1.14 from reviewer 1.
Choosing the cutoff frequency at 0.5 has been also done in previous research. We wanted to follow this approach to generate comparable results. If we understand you correctly, you are suggesting to consider the wave number where the transfer function has dropped to 0.7 as a cut-off frequency? We have compared there these two approaches in the figure below using the model based on Mann turbulence. It shows that choosing the wavenumber where the transfer function dropped to 0.7 will result in a larger wave number and even less filtering.

[Figure]

RC 2.2  Eq. 3 and 4: What is \delta, k'?

AC  \delta is the Dirac delta function. The usage indicates to considering an infinitely small slice of the wavenumber space.

RC 2.3  Eq. 9: Some intermediate steps how to get there might be helpful. Maybe in the appendix? How do they relate to the equations from Mirzaei and Mann (2016)?

AC  We have added more intermediate step in eq. 9.

RC 2.4  Eq. 20 and 21: It is also not really clear, how the correction is applied to the real data, since only 1 Hz data are collected. Is this algorithm done on the lidar system or in post processing? It is also not clear, where this correction comes from. Line-of-sight wind speeds are often used in a wind field reconstruction algorithm which directly provides derived signals such as the REWS. And maybe I am wrong, but the correction seems to be the same than reconstructing the average horizontal wind speed. For small misalignment angles it might be not very important. For larger angles however, it is more the average longitudinal wind having an "effect" on the rotor. Thus it is not clear, why this correction is necessary. But maybe I missed something. Thus, some explication might be helpful.

AC  Yes, this is correct. A correction for turbine misalignment is not necessary. This part has been added by mistake to the manuscript and the misalignment correction has not been used in the data analysis. We have removed this section in the manuscript.

RC 2.5 Organization: The paper is mostly well organized. Only Section 4 might be separated into subsections and Section 2 might be better organized. The part before the current 2.1 could be included in a subsection "Overview Coherence Model". Currently, 2.1 is including the model of the rotor spectrum, 2.2. how you get the REWS estimate from turbine data. Then 2.3 combines model of the lidar spectrum, cross spectrum and how you get the REWS estimate from lidar data. Thus, the subsections seem to be not on the same level. For the understanding, it might be better to first describe the model and its component (2.2 $S\_RR$, 2.3 $S\_LL$, 2.4 $S\_RL$) and then the model implementation and validation against simulation (2.5) and then the modification for field testing (turbine measurements).

AC Yes, we have split up section 4 into three parts and have restructured section 2.

RC 2.6 Eq. 20: Shouldn't \beta be \beta_i?

AC This has been changed in the manuscript.

RC 2.7 p20, l1: Up to this point, it is not mentioned that both lidar systems provides 1 Hz data. Table one might lead to 2 and 4 Hz. So it might be not clear at this point why >90% is equal to 540 measurements).

AC The sampling rate (which is 1 Hz for both systems) has been added to table 1.

RC 2.8 Figure 1: \theta_{FF} and \theta_{FB}. In text on page 1: FF and FB are not in mathmode.

AC Mathmode is now is also used in the text on page 1.

RC 2.9 Captions of Fig. 9 and 10 don't end with a period, others do.

AC This has been changed in the manuscript.

RC 2.10 Figure 3,6,7: unit in labels (partly) missing.

AC The units for the figures 3,6,7 have been added.

**Author Response to Review Comment #3**

Dear Reviewer,

Thank you for reviewing the manuscript. Your comments were very helpful and improved the quality of the manuscript. The author responses can be found below each reviewer comment.

RC 3.1   When comparing the Mann model to the Kaimal model, a more statistical approach would be preferred. For instance, on page 13 line 15, it is stated that ". . . the measured data agrees very well with the Mann turbulence model coherence. The Kaimal model on the other hand seems to give a slight underestimation of coherence". Can you quantify this agreement/underestimation numerically? Perhaps as a function of frequency?

AC   We have added the root-mean-squared-error (RMSE) between measured and modeled coherence to quantify the performance of the model compared to the measured data. A table presenting the RMSE values has been added and a paragraph describing the results.

RC 3.2   Could you provide concisely the benefits of the Mann model over the Kaimal model, and potential drawbacks of the Mann model if applicable? Computation time is mentioned on page 2, lines 13-14, but not elaborated on; and Mann model fitting is mentioned in Appendix C, but it is unclear to me how much more complex the Mann model (3D) is than the Kaimal model (1D) (page 15, line 9).

AC   Just before section 2.1 we have described the pros and cons of the Mann and Kaimal models in some detail and added several additional references.

RC 3.3   In the conclusion, it is stated that blade passage effects were not included. In the experimental data, how was the effect of blade passage removed?

AC   The lidar removes blade interference by default based on the returned spectra, which show a very distinct shape when the laser light is reflected by the blades.

RC 3.4   A fuller description of lidar operation would be preferred, for readers who are not highly familiar with lidar operation. There are terms used that may not be very meaningful to readers, such as Rayleigh length, Doppler spectrum, probe volume, focus distance, and perhaps even wave number.

AC   We have added a small description of the lidar principle at the beginning of sec. 2.3 and extended the description of Rayleigh length, focus distance, Doppler spectrum, probe volume and wave number.

RC 3.5   Is (3) an alternative definition of the spectral tensor? The definition from Mann (1994) does not appear in the manuscript.

AC   (3) is a general definition of any spectral tensor. The definitions in Mann (1994) follow a more strict mathematical description. Here we follow the definitions in Mirzaei and Mann (2016), which are equivalent to Mann (1994).

RC 3.6   I don't understand the i,j indices in (16). Further, the meaning of ". . . summation of repeated indices is implied" (page 6, line 14) is unclear to me. It appears also that in (18) and (19), the i,j indices are replaced with k,l. Could you be explicit about what each of these indices represent (especially since k has been assigned to the wavenumber previously)?

AC   We have expanded the explanation of the indices after (16). We have also added after (19) that the index k should not be confused with the wave number k.

RC 3.7   Labeling on plots needs some improvement. Font sizes are small and labels are unclear. For instance, Figure 6 has "WSP" on the x-axis, but this is not defined.

AC   We have improved the labeling of the figures. "WSP" has been changed to wind speed in figure 6.

RC 3.8   There is inconsistent notation, particularly with Greek lettering. β is used for both pitch angle and azimuth angle, e.g., page 5, line 20 and table 1. β and θ are both used for blade pitch angle, e.g., page 1, line 20 and page 5, line 20. φ is used for both the lidar weighting function (eq. (15)) and the average turbine misalignment (page 7 line 11). alpha is used for both the half-cone opening angle (page 6, line 21) and the Kolmogorov constant (page 11 line 1).

AC   The Greek letter β has been removed from table 1. The Kolmogorov constant is now \alpha_K. The pitch angle is now β in all instances. \varphi is used for the weighting function and \phi for the yaw misalignment.

RC 3.9   It would be nice if some of the key parameters are labeled in the left panel of Fig. 4. For instance, the half-cone angle and the focus distance. The distance d_{Nac} could be labeled in the right panel of Fig. 4, especially that this is not really defined until page 15 (long after table 1 where it first appears). Delta x, which also appears in Table 1, was not completely clear to me until page 15 when it is finally defined.

AC   The illustration has been added to figure 4 and the distance between lidar system and rotor is mentioned in the text.

RC 3.10   Figures 9 and 10 could be labeled more clearly. Perhaps the "Data" line should be labeled "turbine estimates" (or something similar)? Then it may be clearer that each curve represents the coherence between the lidar data and each of the labeled items.

AC   The legend in figures 9 and 10 aims to indicate that here experimental data is compared to two models. The data is not only gathered from the turbine but also the lidar. Thus, we believe that the label "data" is a sufficient distinction from the two models.

RC 3.11 There is inconsistent notation for lidar focus distance ($d_f$ vs $L_f$). See page 6, line 8 and page 7, line 7.

AC The notation has been changed to $d_f$.

RC 3.12 What is the difference between the induction factor and the axial induction factor on page 7? What is the relationship between them?

AC They are the same. "Axial" has been removed on page 7.

RC 3.13 The term 'filter' appears to be used to refer to both a digital filter (in referring to a Butterworth Filter, page 16) and something far more general on page 12. This use (particularly the more general use, on page 12) is confusing in a scientific journal.

AC On page 12, l. 1, it was mentioned that a data filter is applied. The text has been clarified to indicate that this is a filtering of data where measurements are rejected based on certain criteria.

RC 3.14 Readers may find the use of 'region 1, region 2' (page 13 lines 11-12) confusing if coming from a wind energy background, where modes of turbine operation (below cut-in, below rated operation) are also referred to as 'regions'.

AC We have added a note indicating that these regions should not be confused with the operational regions of the wind turbine controller.

RC 3.15 There are some typos and grammatical errors throughout which should be corrected.'

AC We went through the manuscript and corrected the typos and grammatical errors.

[revised manuscript text omitted]

---

## Author Response (AR2)

**Author Response to Review Comment #1 – Part 2**

Dear Reviewer,

Thank you for reviewing the manuscript. Your comments were very helpful and improved the quality of the manuscript. The author responses can be found below each reviewer comment.

| RC 1.1 | RC 1.2 Eqs. 16, 18, 19: Some derivation details (like Eqs. 6-8) or references to sources where these equations are derived would be appreciated. |
|---|---|
| | "AC References to the a paper where details of the derivation of the formulas can be found have been added below the formulas. The reference is also found at the beginning of the section." |
| | Where the new Eq. 22 is discussed, some reference should be pointed out where the derivation details can be found. Otherwise it is not until the next section where a reference is provided. |
| | Pg. 9, ln. 1: "Details of the derivation of the previous two equations can be found in Mirzaei and Mann." The previous two equations are 23 and 24, but I don't see why Eq. 23 needs a reference, as it is quite simple given Eq. 21. |

AC   We have added reference close to the equations 20, 22 and 24.

| RC 1.2 | RC 1.4 Pg. 7, ln. 12: For the yaw misalignment correction, can you explain if you are trying to estimate the total horizontal wind speed, or the component perpendicular to the rotor? Additionally, comment on differences between measurements with the corrected velocities and the spectral model. For example, with yaw misalignment the measured wind will travel toward the rotor at an angle and reach the rotor at a different position than the model assumes. This could cause some differences between the measurements and model that the correction doesn't account for. |
|---|---|
| | "AC Here we trying to estimate the component perpendicular to the rotor. A clarification has been added below eq. 21. Also differences to the model have been pointed out. But in case of small yaw alignment the expected differences are assumed to be small." |
| | It seems that you are removing the proposed correction for yaw misalignment and instead saying that the effects can be ignored for small yaw misalignments. I think this is a fair simplification, but the paragraph needs some work, as it is confusing. The sentence "For example, a turbulent structure will traveling along the mean wind direction…" What is this an example of? Of modeling errors that can exist when ignoring yaw misalignment? I feel that a 2nd sentence in this paragraph is needed, |

explaining that some modeling errors will exist because of effects of yaw misalignment. Then you can list examples before stating that the errors are expected to be small and are ignored.

AC   We have updated the discussion of the model shortcoming. We have also mentioned that for small misalignments the shortcoming is insignificant.

RC 1.3    RC 1.5 Table 2: Do you notice differences in length scales and other parameters if you bin by stability in addition to by sector, and would this be worth considering in the analysis?

"AC We have considered binning by stability as the Mann turbulence model is a representation of turbulence in the neutral atmosphere. However, we saw no deviations of the measured point spectra to the fitted Mann turbulence model for different stability classes, see appendix C. Thus, we did not divide the data into stability classes."

Interesting observation. However, I feel that it should be mentioned in the text briefly that binning by stability did not change the spectra significantly, but that binning by sector showed a large difference. Otherwise, readers might wonder if an opportunity was missed by not binning by stability.

AC   We have added this fact to the text close to table 2.

RC 1.4    RC 1.10 Pg. 15, ln. 7: "When comparing the experimental data to the Kaimal model, a larger mismatch is observed compared to region 1." It appears that the coherence for the Kaimal model does not change much between Figs. 9 and 10, and that the Mann model changes similar to the field measurements. Can you comment on this?

"AC The coherence changes are bigger for the Mann turbulence model because changes in the three dimensional structure of turbulence are better represented by the Mann turbulence model. The Kaimal model is less flexible at representing changes in the three-dimensional structure of the turbulence and thus smaller changes are seen between the two regions for this model."

"The coherence changes are bigger for the Mann turbulence model because changes in the three dimensional structure of turbulence are better represented by the Mann turbulence model." This is a good explanation that clearly states the advantages of the Mann model. I feel that this sentence, or something like it should be added to the text to supplement the existing discussion on pg. 16, lns. 11-15.

AC   We have added a sentence regarding the better representation of 3D turbulence.

RC 1.5    RC 1.12 Pg. 15, ln. 20: "the information theoretical delay estimator" This sounds like a great way to estimate the time delay. Could you briefly clarify how the two input signals are split into past and future? Are you comparing the past part of one signal to the future part of the other?

> "AC The signal is split in the middle. Then both past values of the signal are compared to the both future values. This has been clarified in the manuscript. More details can be found in the reference."
>
> I agree that details should be left for the reference. But the explanation in the text still needs improvement. Consider changing "…into two parts: the past and the future and calculating the mutual information of two signals." to something like: "…into two parts: the past and the future and calculating the mutual information between the past signals and the future signals."

AC   We have added the suggested correction to the text.

RC 1.6    Pg. 7, ln. 23: "The implied sum… could also be written in vector and matrix notation as…" It seems strange to include this since the format of the vector and matrix equation referenced doesn't show up like this in Eq. 20.

AC   We believe that this information could be helpful for the reader. Also a reviewer asked specifically for a clarification (see review comment #3 under point RC 3.6).

RC 1.7    Pg. 9, ln. 3: "the lidar systems probes focus points…" -> "the lidar system probes focus points…"?

AC   This has been corrected.

RC 1.8    Pg. 18, ln. 20: "Still, the results for both lidar systems show great overlap between the measured delay and the delay expected from advection speed and lidar geometry." Especially for the 2-beam lidar, the binned mean values are ~1 second lower than the predictions from the advection speed and lidar geometry for low wind speeds. This should be discussed briefly.

AC   We have added a discussion of this observation.

RC 1.9    Fig. 12: Check the legend for the 4-beam plot.

AC   This has been corrected.

**Author Response to Review Comment #2 – Part2**

Dear Reviewer,

Thank you for reviewing the manuscript. Your comments were very helpful and improved the quality of the manuscript. The author responses can be found below each reviewer comment.

| RC 1.1 | Figure 12 with the delay time analysis is very helpful. Please check the legend of the 4-beam ("Mean bin +/- 1Std" is missing). And you could either write "Time delay $\Delta t$ [s]" in the ylabel as in Equation (25) or simply "Time delay [s]". |
|---|---|

AC   The legend and labels have been modified.

[revised manuscript text omitted]